# TOKEN STATISTICS TRANSFORMER: LINEAR-TIME ATTENTION VIA VARIATIONAL RATE REDUCTION

**Ziyang Wu**     **Tianjiao Ding**[*]     **Yifu Lu**[*]     **Druv Pai**     **Jingyuan Zhang**
UC Berkeley      UPenn               UMich           UC Berkeley      THU & TranscEngram

**Weida Wang**      **Yaodong Yu**      **Yi Ma**                **Benjamin D. Haeffele**
Tsinghua SIGS      UC Berkeley       UC Berkeley & HKU         JHU & UPenn

## ABSTRACT

The attention operator is arguably the key distinguishing factor of transformer architectures, which have demonstrated state-of-the-art performance on a variety of tasks. However, transformer attention operators often impose a significant computational burden, with the computational complexity scaling quadratically with the number of tokens. In this work, we propose a novel transformer attention operator whose computational complexity scales linearly with the number of tokens. We derive our network architecture by extending prior work which has shown that a transformer style architecture naturally arises by "white-box" architecture design, where each layer of the network is designed to implement an incremental optimization step of a maximal coding rate reduction objective (MCR$^2$). Specifically, we derive a novel variational form of the MCR$^2$ objective and show that the architecture that results from unrolled gradient descent of this variational objective leads to a new attention module called Token Statistics Self-Attention (`TSSA`). `TSSA` has *linear computational and memory complexity* and radically departs from the typical attention architecture that computes pairwise similarities between tokens. Experiments on vision, language, and long sequence tasks show that simply swapping `TSSA` for standard self-attention, which we refer to as the Token Statistics Transformer (ToST), achieves competitive performance with conventional transformers while being significantly more computationally efficient and interpretable. Our results also somewhat call into question the conventional wisdom that pairwise similarity style attention mechanisms are critical to the success of transformer architectures. Code is available at `https://github.com/RobinWu218/ToST`.

## 1 MOTIVATION

Transformer architectures have led to state-of-the-art performance across many applications in machine learning, computer vision, natural language processing, and elsewhere (Vaswani et al., 2017; Devlin et al., 2019; Radford et al., 2018; Chen et al., 2020; Dosovitskiy et al., 2020). Arguably, the defining component of transformers is the *attention operator*, which was originally motivated to allow for handling long-range interactions among data tokens (e.g., image patches, words, video frames). Attention can come in a variety of forms, but transformer architectures typically employ *self-attention* (Vaswani et al., 2017). In particular, the core aspect of the self-attention operator is *scaled dot product attention*, which generates output tokens as weighted averages of input tokens, with weights computed as a "similarity" between pairs of input tokens, i.e., the scaled dot product between pairs of tokens after multiplication by the "key" and "query" parameter matrices.

While the success of transformers (and by extension self-attention) in a wide variety of applications demonstrates the utility of the approach, it comes with a potentially significant drawback: it requires one to compute a similarity between all pairs of input tokens, which results in its *computation and memory complexity scaling quadratically* with the number of input tokens. Indeed, the computational demands of self-attention operators are well-noted in the literature, and a number of

---
[*]Equal contribution

techniques have been proposed to allow for more efficient computation[1]. Examples include partitioning the tokens into blocks and having each attention head only compute with one subset of the tokens (Qiu et al., 2020), computing attention over various sliding windows of tokens (Beltagy et al., 2020; Liu et al., 2021), finding a suitable low-rank projection of the tokens (Wang et al., 2020), or approximating the pairwise attention computation via the Nystrom extension (Xiong et al., 2021). Likewise, Ali et al. (2021) propose to calculate scaled dot products between feature channels rather than between tokens, noting relationships between the eigenspaces of Gram matrices and correlation matrices, and dub their method the cross-covariance transformer (XCiT). In this paper, we will also propose an efficient attention operator, but we will derive our method from a radically different approach which suggests that *computing (or approximating) pairwise similarities between tokens may not be necessary at all*. To build an intuition for this notion it is useful to consider the core operation of the attention operator, which has been noted by multiple authors in the literature (Chaudhari et al., 2021; Zhang et al., 2021; Vidal, 2022) as performing a form of kernel regression (Nadaraya, 1964; Watson, 1964) with close connections to classical denoising techniques such as non-local means or block-matching and 3D filtering (BM3D) (Buades et al., 2005; Dabov et al., 2006). Namely, output tokens are formed by taking a weighted average of input tokens which are deemed to be "similar" (as measured by the learned scaled dot product metric). More abstractly, this suggests that the self-attention operator is a particular case of a more general class of operators which produces outputs based on statistics computed from the input tokens. In other words, in the case of self-attention we compute weighted averages based on the learned scaled dot product metric, but one could envision producing outputs based on other statistics of the input tokens which are more efficiently computed.

Here, to derive our proposed method we will leverage "white-box" architecture design (also referred to as *algorithmic unrolling*), which takes the philosophy that the operations of the layers of a neural network can be interpreted as performing incremental updates to optimize an objective function (Gregor & LeCun (2010); cf. Monga et al. (2021) and its references). In particular, the structure of the architecture is defined by the form of an update that a specific opti-

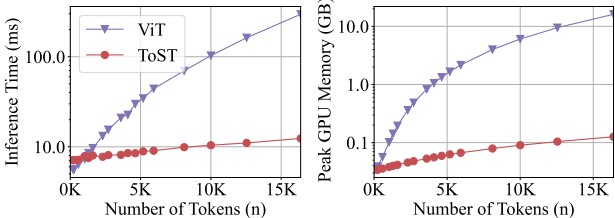

Figure 1: Our TOST architecture, from unrolling a novel variational form of MCR$^2$, **is faster and uses less memory** than standard transformer architectures such as ViT (note the log-scale y-axes) and is based on a dramatically different notion of attention.

mization method (e.g., gradient descent/ascent) employs, but the parameters of the resulting update are still left to be learned[2]. Notably, recent work (Yu et al., 2023; 2024) has argued that transformer-style architectures arise naturally from white-box design when incrementally updating an objective function based on the principle of maximum coding rate reduction (MCR$^2$). While this provides interesting interpretations of transformer architectures, the resulting white-box architecture is largely identical to a transformer, so it still suffers from the same computational challenges described above. Nevertheless, here we draw inspiration from this prior work and make the following contributions:

1. We provide a novel variational form for the MCR$^2$ objective that additionally allows one to upper-bound certain spectral functions of large matrices, which may be of independent interest.

2. Using white-box architecture design to derive an incremental optimization of our variational objective, we propose a novel attention operator which is highly efficient and only requires *linear complexity* to compute (see Figure 1 comparing our method with ViT (Dosovitskiy et al., 2020)).

3. Our resulting architecture is radically different from standard transformers in that it does not compute/approximate pairwise similarities between input tokens. Instead, it only computes a data-dependent low-rank projection based on an empirical second moment statistic of the input token features. This leads us to name it the *Token Statistics Transformer* (TOST) (see Figure 3).

4. We validate our approach experimentally and show that our proposed architecture achieves comparable performance to standard transformer architectures while having a significantly reduced computation and memory footprint, particularly for large numbers of high-dimensional tokens.

---

[1]See App. B.1 for discussion on related efficient models such as linear attention and state-space models.

[2]Note that the objective function used to derive the architecture and the training objective used to learn the model parameters need not be the same.

Notably, replacing self-attention with our attention operator typically at least maintains performance on benchmark tasks and often improves performance for tasks with large token lengths.

## 2 BACKGROUND AND PRELIMINARIES

**Notation.** We define a few notations that will be used throughout the rest of the paper. For a positive integer $n$, let $[n] \doteq \{1, 2, \ldots, n\}$. Let $\mathbf{0}$ (resp. $\mathbf{1}$) be the vector/matrix of all zeros (resp. all ones). Let $\mathbf{I}$ be the identity matrix. For $\mathbf{0}$, $\mathbf{1}$, and $\mathbf{I}$, the dimension is inferred unambiguously from context. For a matrix $\mathbf{A} \in \mathbb{R}^{m \times n}$, we denote by $\mathbf{a}_i \in \mathbb{R}^m$ the $i^{\text{th}}$ column of $\mathbf{A}$, and by $A_{ij} \in \mathbb{R}$ the $(i, j)^{\text{th}}$ entry of $\mathbf{A}$. For a vector $\mathbf{v} \in \mathbb{R}^n$, let $\text{Diag}(\mathbf{v}) \in \mathbb{R}^{n \times n}$ be a diagonal matrix with the entries of $\mathbf{v}$ along the diagonal. For a matrix $\mathbf{A} \in \mathbb{R}^{m \times n}$ we denote by $\mathbf{A}^{\odot 2} \in \mathbb{R}^{m \times n}$ the element-wise square of $\mathbf{A}$. For a vector $\mathbf{v} \in \mathbb{R}^n$ and a function $f \colon \mathbb{R} \to \mathbb{R}$, let $f[\mathbf{v}] \in \mathbb{R}^n$ be the element-wise application of $f$ to the entries of $\mathbf{v}$. For $m \geq n$, we denote by $\text{O}(m, n) \subseteq \mathbb{R}^{m \times n}$ the set of $m \times n$ matrices with orthonormal columns; consequently, we denote by $\text{O}(n) \doteq \text{O}(n, n)$ the set of $n \times n$ orthogonal matrices. We denote by $\text{PSD}(n) \subseteq \mathbb{R}^{n \times n}$ the set of $n \times n$ positive semi-definite (PSD) matrices. For $\mathbf{M} \in \text{PSD}(n)$ and $i \in [n]$, we denote by $\lambda_i(\mathbf{M})$ the $i^{\text{th}}$ largest eigenvalue of $\mathbf{M}$.

### 2.1 REPRESENTATION LEARNING VIA MAXIMAL CODING RATE REDUCTION

Before presenting the development of our attention operator, we first review core ideas that lead to its derivation. Real data, such as images, videos, and text, are often assumed to be realizations of some random variable $\mathbf{X}$ drawn from some high-dimensional probability distribution which has an underlying rich low-dimensional structure (Fefferman et al., 2016; Wright & Ma, 2022). Commonly, preprocessing methods *tokenize* the data, converting each sample into a list of vectors (i.e. tokens) which typically represent a small and local part of the data (e.g., a word or piece of a word, a patch of an image). As a result, for applications involving transformers, $\mathbf{X} = [\mathbf{x}_1, \ldots, \mathbf{x}_n] \in \mathbb{R}^{D \times n}$ is typically a matrix-valued variable, and the transformer seeks to find a *representation mapping*, or a mapping of the tokens $\phi \colon \mathbb{R}^{D \times n} \to \mathbb{R}^{d \times n}$ such that the features $\mathbf{Z} = \phi(\mathbf{X}) = [\mathbf{z}_1, \ldots, \mathbf{z}_n] \in \mathbb{R}^{d \times n}$ are appropriate for the task at hand[3]. While many works focus primarily on how well $\mathbf{Z}$ can be used to solve a particular task (e.g., classification or segmentation), recent work (Yu et al., 2020; Chan et al., 2022; Ding et al., 2023; Chu et al., 2024) has proposed a more task-agnostic approach to learning a suitable representation which, rather than focusing on a task-specific metric, instead seeks to find the underlying low-dimensional structure in the data via an objective based on the principle of maximal coding rate reduction (MCR$^2$). More specifically, the tokens are assumed to belong to an underlying set of $K$ groups, which may (for example) represent different modes within the population of tokens, and the MCR$^2$ objective seeks to find token features of each group which are *compressed* within a group while the set of all token features is as expansive as possible. In particular, let $\mathbf{\Pi} = [\boldsymbol{\pi}_1, \ldots, \boldsymbol{\pi}_K] \in \mathbb{R}^{n \times K}$ be a stochastic "group assignment" matrix (i.e., $\mathbf{\Pi 1} = \mathbf{1}$ and $\Pi_{ik} \geq 0, \; \forall (i, k) \in [n] \times [K]$) that assigns a group membership probability to each token (i.e., the $i^{\text{th}}$ row of $\mathbf{\Pi}$ is a probability vector for group membership for the $i^{\text{th}}$ token). Then, letting $\epsilon > 0$ and $n_k \doteq \langle \boldsymbol{\pi}_k, \mathbf{1} \rangle$ for each $k \in [K]$, the MCR$^2$ objective, denoted as $\Delta R(\mathbf{Z}, \mathbf{\Pi})$, is:

$$\Delta R(\mathbf{Z}, \mathbf{\Pi}) \doteq \underbrace{\frac{1}{2} \log \det \left( \mathbf{I} + \frac{d}{\epsilon^2} \frac{1}{n} \mathbf{Z Z}^\top \right)}_{\doteq R(\mathbf{Z})} - \underbrace{\frac{1}{2} \sum_{k=1}^{K} \frac{n_k}{n} \log \det \left( \mathbf{I} + \frac{d}{\epsilon^2} \frac{1}{n_k} \mathbf{Z} \text{Diag}(\boldsymbol{\pi}_k) \mathbf{Z}^\top \right)}_{\doteq R_c(\mathbf{Z}, \mathbf{\Pi})}. \quad (1)$$

Intuitively, the $R(\mathbf{Z})$ term measures the volume of all the features, so maximizing it w.r.t. $\mathbf{Z}$ would promote that the features spread out (hence, we call it the "expansion term"). The $R_c(\mathbf{Z}, \mathbf{\Pi})$ term measures the sum of volumes of features from each of the $K$ groups encoded by $\mathbf{\Pi}$, so minimizing it (or maximizing $-R_c$) w.r.t. $\mathbf{Z}$ would push features in each group to be close (hence we call it the "compression term"). (Yu et al., 2020) showed that learning a representation by maximizing equation 1 under suitable normalization constraints leads to the features being *compressed*, *linearized*, and *discriminative*: features from the same group lie in a low-dimensional subspace corresponding to the intrinsic low-dimensionality of the token data, and the subspaces they form are orthogonal to each other. See Figure 2 for a visualization.

---

[3]Here, with some abuse of notation, we will use $\mathbf{Z}$ to refer to the the token features at some intermediate layer (or point) within the network, not necessarily just the final features at the output of the network. We also pack the tokens into the columns of a matrix, rather than the rows as is often done in the transformer literature, as we believe it leads to a cleaner presentation.

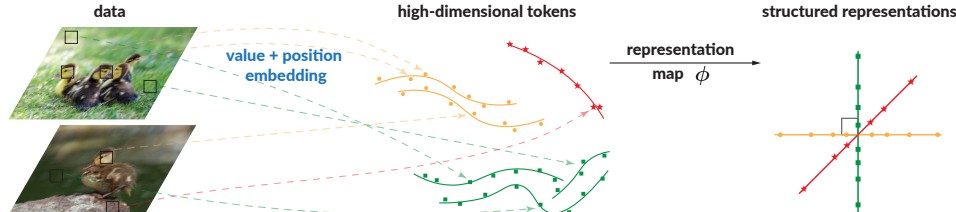

Figure 2: **Tokenization and representation for image data.** Data (images) are split up into tokens (patches) $\mathbf{X}$, which share semantics with other tokens from the same or different samples. Tokens with similar semantics may belong to the same geometric structures in the original space and be grouped together by $\mathbf{\Pi}$. A learned mapping $\phi$ converts these tokens into features which are compressed, linearized, and discriminative.

## 2.2 BUILDING WHITE-BOX DEEP NETWORKS VIA UNROLLED OPTIMIZATION

Having introduced the MCR$^2$ objective, we now discuss the principle of designing network architectures in a "white-box" manner (i.e., where the structure of the architecture is well motivated and interpretable) via *algorithmic unrolling* (Gregor & LeCun (2010); Yu et al. (2024); see also the review Monga et al. (2021) and references therein). Algorithmic unrolling constructs each layer of the network to (approximately) implement a step of an optimization algorithm for our representation learning objective, where the network architecture is defined by the structure of an update step from an optimization algorithm (e.g., gradient ascent/descent) but the parameters of the update are not fixed at their analytical values (e.g., the computed gradient for a gradient step) but parameterized and learned through model training. Notably, Yu et al. (2024) argue that by unrolling a slightly modified MCR$^2$ objective, which additionally adds a $\ell^0$ sparsity penalty on $\mathbf{Z}$ to promote axis-aligned solutions, and uses a modified compression term $\bar{R}_c(\mathbf{Z})$, one arrives at an architecture which is almost identical to a standard transformer (with the primary difference being the query, key, and value parameter matrices are all equal in the self-attention operator). In particular, a gradient step on the compression term, i.e., $\mathbf{Z}^+ = \mathbf{Z} - \tau\nabla_\mathbf{Z}\bar{R}_c(\mathbf{Z})$, approximates a *multi-head self-attention* operator with weight sharing, plus a residual connection. That is, *incremental optimization of the compression term yields a self-attention operator.* Further, a proximal gradient step on the remaining terms $R(\mathbf{Z}) - \|\mathbf{Z}\|_0$ gives a soft thresholding operation which has a similar functional form to a two-layer perceptron with ReLU activation. Taken together, these resulting operations closely approximate a layer of a standard transformer (self-attention followed by a two-layer perceptron). The transformer-like network constructed by concatenating these layers is named CRATE, and performs comparably to the empirically engineered ViT architecture (Dosovitskiy et al., 2020). However, as the attention mechanism in CRATE is largely identical to standard self-attention, it still has time and memory complexity that scales *quadratically* with $n$, the number of tokens. This quadratic complexity presents a large barrier to scaling up transformer-like models both within and without the CRATE paradigm. In the sequel, inspired by how CRATE derives a self attention-like operation as a gradient step on a compression term similar to $R_c$, we will construct a new attention mechanism as a gradient step on a novel *variational reformulation* of $R_c$.

## 3 PROPOSED ATTENTION FORMULATION

We now describe our new proposed attention mechanism, which we will derive by unrolled optimization of a novel variational form of the compression term within the MCR$^2$ objective.

### 3.1 A NEW VARIATIONAL FORM FOR CODING RATES

To begin, we consider a general form of MCR$^2$-like objectives based on concave functions of the spectrum of a matrix. Namely, for a given PSD matrix $\mathbf{M} \in \mathrm{PSD}(d)$ and any scalar $c \geq 0$ we have that $\log\det(\mathbf{I}+c\mathbf{M}) = \sum_{i=1}^d \log(1+c\lambda_i(\mathbf{M}))$ where (recall) $\lambda_i(\mathbf{M})$ is the $i^{\mathrm{th}}$ largest eigenvalue of $\mathbf{M}$. Further note that $\log(1 + c\sigma)$ is a concave non-decreasing function of $\sigma$. Thus, we describe our results in terms of a more general form of MCR$^2$ based on general spectral functions of PSD matrices of the form $F(\mathbf{M}) = \sum_{i=1}^d f(\lambda_i(\mathbf{M}))$, where $f$ is concave and non-decreasing. In particular, recall from our above discussion that the attention mechanism arises from unrolling the compression component of MCR$^2$, so we consider a more general MCR$^2$-style compression function:

$$R_{c,f}(\mathbf{Z}, \mathbf{\Pi}) \doteq \frac{1}{2} \sum_{k=1}^{K} \frac{n_k}{n} F\left(\frac{1}{n_k} \mathbf{Z}\mathrm{Diag}(\boldsymbol{\pi}_k)\mathbf{Z}^\top\right). \tag{2}$$

For objectives of this form, we now note the following result:

**Theorem 1.** *Let $f\colon [0,\infty) \to \mathbb{R}$ be non-decreasing, concave, and obey $f(0) = 0$, and let $F\colon \mathrm{PSD}(d) \to \mathbb{R}$ have the form $F(\mathbf{M}) = \sum_{i=1}^{d} f(\lambda_i(\mathbf{M}))$. Then for each $\mathbf{M} \in \mathrm{PSD}(d)$ and $\mathbf{Q} \in \mathrm{O}(d)$, we have*

$$F(\mathbf{M}) \leq \sum_{i=1}^{d} f\left((\mathbf{Q}^\top \mathbf{M}\mathbf{Q})_{ii}\right). \tag{3}$$

*Further, the inequality in equation 3 is achieved with equality for any $\mathbf{Q}$ which diagonalizes $\mathbf{M}$, and if $f$ is strictly concave then the inequality in equation 3 is achieved with equality if and only if $\mathbf{Q}$ diagonalizes $\mathbf{M}$.*

Theorem 1 is a straightforward corollary of Theorem 2, which is stated and proved in Appendix A. Theorem 1 allows us to upper-bound a function of large matrices via just computing a scalar function of the *diagonals* of a matrix product, and if we know the spectral structure of the input matrix then we can craft orthogonal matrices for which this upper bound is made tight. This is a powerful tool and may be of independent interest.

## 3.2 Efficient Deep Architecture via Unrolling the Variational Form

Using the above result, we can replace equation 2 with an equivalent variational objective with form

$$R_{c,f}^{\mathrm{var}}(\mathbf{Z}, \mathbf{\Pi} \mid \{\mathbf{U}_k\}_{k=1}^{K}) \doteq \frac{1}{2} \sum_{k=1}^{K} \frac{n_k}{n} \sum_{i=1}^{d} f\left(\frac{1}{n_k}(\mathbf{U}_k^\top \mathbf{Z}\mathrm{Diag}(\boldsymbol{\pi}_k)\mathbf{Z}^\top \mathbf{U}_k)_{ii}\right), \tag{4}$$

where the equivalence is in the sense that for an optimal choice of $\{\mathbf{U}_k \in \mathrm{O}(d)\}_{k=1}^{K}$ matrices as described in Theorem 1 (i.e., orthogonal matrices which diagonalize each $\mathbf{Z}\mathrm{Diag}(\boldsymbol{\pi}_k)\mathbf{Z}^\top$) we will achieve a tight bound with $R_{c,f}^{\mathrm{var}}(\mathbf{Z}, \mathbf{\Pi} \mid \{\mathbf{U}_k\}_{k=1}^{K}) = R_{c,f}(\mathbf{Z}, \mathbf{\Pi})$. Note that in general, achieving this bound would require selecting, for each sampled instance of $\mathbf{Z}$, a new optimal set of $\mathbf{U}_k$ parameter matrices which diagonalize each $\mathbf{Z}\mathrm{Diag}(\boldsymbol{\pi}_k)\mathbf{Z}^\top$, which is clearly impractical for a network architecture. Instead, as an alternative viewpoint, rather than considering the data ($\mathbf{Z}$) as fixed and trying to optimize the $\mathbf{U}_k$ parameters to achieve the tight variational bound, we can instead take the algorithmic unrolling design principle described above and design an operator to perturb $\mathbf{Z}$ to incrementally minimize $R_{c,f}^{\mathrm{var}}(\cdot \mid \{\mathbf{U}_k\}_{k=1}^{K})$. To make this point explicit, each variational bound becomes tight when the eigenspaces of $\mathbf{Z}\mathrm{Diag}(\boldsymbol{\pi}_k)\mathbf{Z}^\top$ align with the columns of $\mathbf{U}_k$, so by rotating the appropriate columns of $\mathbf{Z}$ (namely, those which correspond to large entries in $\boldsymbol{\pi}_k$) to align with $\mathbf{U}_k$ we can approach a tight variational bound. That is, instead of rotating $\mathbf{U}_k$ to align with the data for each instance of $\mathbf{Z}$, we can instead rotate the token features in each $\mathbf{Z}$ to align with $\mathbf{U}_k$.

Following this approach, we compute a gradient descent step on $R_{c,f}^{\mathrm{var}}$ w.r.t. $\mathbf{Z}$. To begin this computation, first let $\boldsymbol{\pi} \in \mathbb{R}^n$ be any element-wise non-negative vector. Then we have

$$\nabla_{\mathbf{Z}} \frac{1}{2} \sum_{i=1}^{d} f((\mathbf{Z}\mathrm{Diag}(\boldsymbol{\pi})\mathbf{Z}^\top)_{ii}) = \mathrm{Diag}(\nabla f[\mathbf{Z}^{\odot 2}\boldsymbol{\pi}])\mathbf{Z}\mathrm{Diag}(\boldsymbol{\pi}), \tag{5}$$

where $\nabla f$ is the gradient of $f$, and (recall) $\nabla f[\cdot]$ applies $\nabla f$ to each element of the vector in the bracket. In particular, for $f(x) = \log(1 + (d/\epsilon^2)x)$, $\nabla f(x) = (d/\epsilon^2)(1 + (d/\epsilon^2)x)^{-1}$ is simply a non-linear activation. Also, (recall) $n_k = \langle \boldsymbol{\pi}_k, \mathbf{1} \rangle$. Thus, the gradient of $R_{c,f}^{\mathrm{var}}$ w.r.t. $\mathbf{Z}$ is:

$$\nabla_{\mathbf{Z}} R_{c,f}^{\mathrm{var}}(\mathbf{Z}, \mathbf{\Pi} \mid \{\mathbf{U}_k\}_{k=1}^{K}) = \frac{1}{n} \sum_{k=1}^{K} \mathbf{U}_k \underbrace{\mathrm{Diag}\left(\nabla f\left[(\mathbf{U}_k^\top \mathbf{Z})^{\odot 2} \frac{\boldsymbol{\pi}_k}{\langle \boldsymbol{\pi}_k, \mathbf{1} \rangle}\right]\right)}_{\doteq \mathbf{D}(\mathbf{Z}, \boldsymbol{\pi}_k \mid \mathbf{U}_k)} \mathbf{U}_k^\top \mathbf{Z}\mathrm{Diag}(\boldsymbol{\pi}_k). \tag{6}$$

(Note that the $1/n$ constant arises from a $(n_k/n) \cdot (1/n_k) = 1/n$ constant in each term of the sum.) If we now consider a gradient step w.r.t. the $j^{\text{th}}$ token $\mathbf{z}_j$, we arrive at our proposed incremental compression operator, i.e., our surrogate for a *self attention* + residual operator:

$$\mathbf{z}_j^+ = \mathbf{z}_j - \tau \nabla_{\mathbf{z}_j} R_{c,f}^{\mathrm{var}}(\mathbf{Z}, \mathbf{\Pi} \mid \{\mathbf{U}_k\}_{k=1}^{K}) = \mathbf{z}_j - \frac{\tau}{n} \sum_{k=1}^{K} \Pi_{jk} \mathbf{U}_k \mathbf{D}(\mathbf{Z}, \boldsymbol{\pi}_k \mid \mathbf{U}_k)\mathbf{U}_k^\top \mathbf{z}_j \tag{7}$$

for each $j \in [n]$, where $\tau > 0$ is a step size parameter for the incremental optimization.

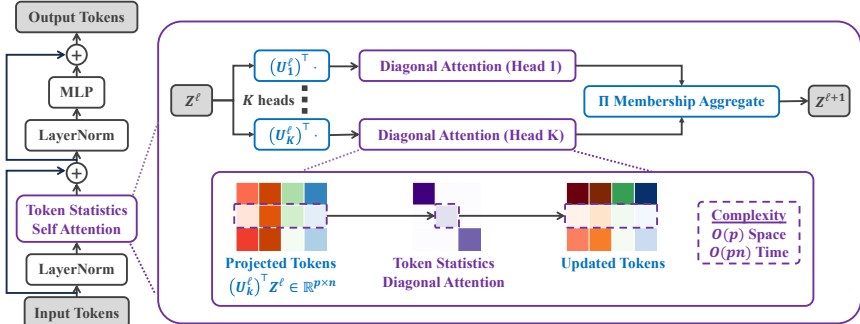

**Figure 3: One layer $\ell$ of the proposed Token Statistics Transformer (TOST).** Notably, the self-attention of TOST transforms tokens $\mathbf{Z}^\ell$ efficiently to $\mathbf{Z}^{\ell+1}$, via multiplying each row of the projected token by *only a scalar*. This leads to reduced complexity of the attention (cf. Table 1): it has $\mathcal{O}(p)$ space and $\mathcal{O}(pn)$ time complexity, where $p$ is the dimension of the projected tokens of each head, and $n$ is the number of tokens.

**Model interpretation.** Given the proposed attention operator in equation 7, first recall that the rows of $\mathbf{\Pi}$ are non-negative and sum to 1, so our operator takes a weighted average of $K$ "attention head"-esque operators and then adds a residual connection. Using that $\sum_{k=1}^K \Pi_{jk} = 1$, we can rewrite equation 7 as:

$$\mathbf{z}_j^+ = \sum_{k=1}^K \Pi_{jk} \Big[ \mathbf{z}_j \underbrace{- \frac{\tau}{n} \mathbf{U}_k \mathbf{D}(\mathbf{Z}, \boldsymbol{\pi}_k \mid \mathbf{U}_k) \mathbf{U}_k^\top \mathbf{z}_j}_{\text{action of one attention head}} \Big]. \tag{8}$$

That is, we can view each attention head as first projecting the token features onto the basis $\mathbf{U}_k$ via multiplying by $\mathbf{U}_k^\top$, multiplying by the diagonal matrix $\mathbf{D}(\mathbf{Z}, \boldsymbol{\pi}_k \mid \mathbf{U}_k)$ (abbreviated as $\mathbf{D}_k$), projecting back into the standard basis via multiplying by $\mathbf{U}_k$, and subtracting this from the original token features via the residual connection. The core aspect of our attention layer is the computation of $\mathbf{D}_k$. Namely, $\Pi_{jk} \geq 0$, so $\boldsymbol{\pi}_k / \langle \boldsymbol{\pi}_k, \mathbf{1} \rangle \in \mathbb{R}^n$ forms a probability distribution over which tokens belong to the $k^{\text{th}}$ group. As a result, $(\mathbf{U}_k^\top \mathbf{Z})^{\odot 2} \boldsymbol{\pi}_k / \langle \boldsymbol{\pi}_k, \mathbf{1} \rangle$ estimates the second moment of $\mathbf{U}_k^\top \mathbf{Z}$ under the distribution given by $\boldsymbol{\pi}_k / \langle \boldsymbol{\pi}_k, \mathbf{1} \rangle$. Further, since $f$ is a concave non-decreasing function, $\nabla f(x)$ monotonically decreases towards 0 as $x$ increases, so the entries of $\mathbf{D}_k$ (which have form $\nabla f(x)$) achieve their maximum at $x = 0$ and decay monotonically to 0 as $x$ increases.

From this, we arrive at the core interpretation of our attention head + residual operators $[\mathbf{I} - (\tau/n)\mathbf{U}_k \mathbf{D}_k \mathbf{U}_k^\top]$. Namely, this operator does an approximate low-rank data-dependent projection, where directions which have a large amount of "power" after the projection $\mathbf{U}_k^\top \mathbf{Z}$ (i.e., directions which have a large second moment $(\mathbf{U}_k^\top \mathbf{Z})^{\odot 2} \boldsymbol{\pi}_k / \langle \boldsymbol{\pi}_k, \mathbf{1} \rangle$) are preserved, while directions which do not are suppressed. To see this, recall that the entries of $\mathbf{D}_k$ decrease monotonically to 0 as the second moment increases, so for directions with large second moments the attention + residual operator acts largely as the identity operator. Conversely, for directions with a small second moment our operator subtracts a projection of the tokens along those directions, resulting in those directions being suppressed. Compared to the standard self-attention operator, our method clearly does not compute any pairwise similarities between tokens. Rather, the interactions between the tokens in $\mathbf{Z}$ impact the operator solely through their contribution to the second moment statistic used to construct the $\mathbf{D}_k$'s. Nevertheless, similar to the standard self-attention operator, our method still has a clear interpretation as performing a form of compression towards a data-dependent low-rank structure, in the sense that it performs an approximate low-rank projection, where the specific directions which are suppressed are those which are not strongly aligned with other tokens in the group.

## 3.3 PRACTICAL IMPLEMENTATION CONSIDERATIONS

Having introduced our proposed attention operator, we now discuss further practical considerations. First, until this point in the presentation we have avoided discussion of how tokens are "grouped" into various attention heads via the $\mathbf{\Pi}$ matrix, but clearly a means of constructing $\mathbf{\Pi}$ is needed to implement our method. Additionally, our variational form in Theorem 1 requires the $\mathbf{U}$ matrices to be square and orthogonal, but one would ideally like to use smaller matrices (i.e., reduce the number of columns in $\mathbf{U}$) for efficiency as well as drop the orthogonality constraints.

Table 1: **Time and space complexity for attention operators in different transformer architectures**: ViT (Dosovitskiy et al., 2020), CRATE (Yu et al., 2024), XCiT (Ali et al., 2021), and the proposed TOST.

|  | ViT | CRATE | XCiT | TOST (ours) |
|---|---|---|---|---|
| Compute time complexity | $\mathcal{O}(pn^2)$ | $\mathcal{O}(pn^2)$ | $\mathcal{O}(p^2n)$ | $\mathcal{O}(pn)$ |
| Memory complexity | $\mathcal{O}(n^2)$ | $\mathcal{O}(n^2)$ | $\mathcal{O}(p^2)$ | $\mathcal{O}(p)$ |

In practice, we do not enforce the orthogonality constraints. To reduce the number of columns in the $\mathbf{U}$ matrices, we note that similar to CRATE (Yu et al., 2024), if we assume the features $\mathbf{Z}$ within group $k$ are (approximately) clustered around a low-dimensional subspace — say of dimension $p$ — then the within-group-$k$ covariance $\mathbf{Z}\mathrm{Diag}(\boldsymbol{\pi}_k)\mathbf{Z}^\top$ is low-rank, where recall that Yu et al. (2020) shows that the optimal geometry of $\mathbf{Z}$ will be for each group to be a low-rank subspace, orthogonal to the other groups. We can thus explicitly find a low-dimensional orthonormal basis for the image of this covariance, i.e., the linear span of the data in group $k$. If the dimension is $p \leq d$, the basis can be represented by a $d \times p$ orthogonal matrix $\mathbf{U}_k \in \mathrm{O}(d, p)$. In this case, we can more efficiently upper-bound $R_c$ using these low-rank orthogonal basis matrices. To show this, we use a more general version of Theorem 1 (see Theorem 2 in Appendix A) to yield the following corollary.

**Corollary 1.** *Let* $f\colon [0, \infty) \to \mathbb{R}$ *be non-decreasing, concave, and obey* $f(0) = 0$, *and let* $F\colon \mathrm{PSD}(p) \to \mathbb{R}$ *have the form* $F(\mathbf{M}) = \sum_{i=1}^p f(\lambda_i(\mathbf{M}))$. *Let* $\mathbf{Z}, \boldsymbol{\Pi}$ *be fixed. Then, for all* $\mathbf{U}_1, \ldots, \mathbf{U}_K \in \mathrm{O}(d, p)$ *such that* $\mathrm{image}(\mathbf{Z}\mathrm{Diag}(\boldsymbol{\pi}_k)\mathbf{Z}^\top) \subseteq \mathrm{image}(\mathbf{U}_k)$ *for all* $k \in [K]$, *we have*

$$R_{c,f}(\mathbf{Z}, \boldsymbol{\Pi}) \leq R_{c,f}^{\mathrm{var}}(\mathbf{Z}, \boldsymbol{\Pi} \mid \{\mathbf{U}_k\}_{k=1}^K), \tag{9}$$

*where* $R_{c,f}^{\mathrm{var}}$ *is formally defined in equation 4. Equality holds if* $\mathbf{U}_k$ *diagonalizes* $\mathbf{Z}\mathrm{Diag}(\boldsymbol{\pi}_k)\mathbf{Z}^\top$ *for each* $k \in [K]$, *and if* $f$ *is strongly concave then this equality condition becomes an "if and only if."*

The final step to define our attention operator is to estimate the group membership $\boldsymbol{\Pi}$. For this we posit a simple model of how each feature $\mathbf{z}_j$ deviates from its supporting subspace and then find the optimal subspace assignment. Yu et al. (2024) show that if we independently model each $\mathbf{z}_j$ as belonging to a low-dimensional Gaussian mixture model, where each Gaussian has a covariance matrix with identical spectrum and is supported on a subspace with orthonormal basis $\mathbf{U}_k$, plus independent Gaussian noise with covariance $\eta\mathbf{I}$, then the posterior probability that each token $\mathbf{z}_j$ belongs to each subspace is given by the assignment matrix $\boldsymbol{\Pi} = \boldsymbol{\Pi}(\mathbf{Z} \mid \{\mathbf{U}_k\}_{k=1}^K)$ as follows:

$$\boldsymbol{\Pi} = \begin{bmatrix} \boldsymbol{\nu}(\mathbf{z}_1 \mid \{\mathbf{U}_k\}_{k=1}^K)^\top \\ \vdots \\ \boldsymbol{\nu}(\mathbf{z}_n \mid \{\mathbf{U}_k\}_{k=1}^K)^\top \end{bmatrix}, \quad \text{where} \quad \boldsymbol{\nu}(\mathbf{z}_j \mid \{\mathbf{U}_k\}_{k=1}^K) \doteq \mathrm{softmax}\left(\frac{1}{2\eta}\begin{bmatrix} \|\mathbf{U}_1^\top\mathbf{z}_j\|_2^2 \\ \vdots \\ \|\mathbf{U}_K^\top\mathbf{z}_j\|_2^2 \end{bmatrix}\right), \quad \forall j \in [n], \tag{10}$$

where $\eta$ becomes a learnable temperature parameter. Thus, given an input feature $\mathbf{Z}$, we estimate $\boldsymbol{\Pi}$ using equation 10 then compute the attention operator. Combining the construction of $\boldsymbol{\Pi}$ in equation 10 with equation 7, we obtain the *Token Statistics Self-Attention* (TSSA) operator:

$$\mathrm{TSSA}(\mathbf{Z} \mid \{\mathbf{U}_k\}_{k=1}^K) \doteq -\frac{\tau}{n}\sum_{k=1}^K \mathbf{U}_k\mathbf{D}(\mathbf{Z}, \boldsymbol{\pi}_k \mid \mathbf{U}_k)\mathbf{U}_k^\top\mathbf{Z}\mathrm{Diag}(\boldsymbol{\pi}_k), \tag{11}$$

where $\boldsymbol{\pi}_k$ are the columns of $\boldsymbol{\Pi} = \boldsymbol{\Pi}(\mathbf{Z} \mid \{\mathbf{U}_k\}_{k=1}^K)$, defined in equation 10, and $\mathbf{D}$ is defined in equation 6. We use this TSSA operator to construct an architecture, the *Token Statistics Transformer* (TOST). Namely, our focus is to construct an alternative attention mechanism, so to obtain TOST we replace the attention operator in XCiT (Ali et al., 2021) with the TSSA operator (with $f(x) = \log(1 + (d/\epsilon^2)x)$ as in Section 3.1), along with other changes detailed in Section 4 which serve to simplify the architecture. We visualize one layer of TOST in Figure 3, and provide pseudocode for the model in Appendix E.

**Complexity analysis.** Table 1 gives a comparison of computational complexity between each attention head of the TSSA operator in our proposed TOST architecture and those of attention mechanisms in other existing transformers. As we see, the TSSA attention head is asymptotically less complex than alternatives in either $n$ or $p$ for both compute time and memory requirements. This is confirmed in real computation metrics shown in Figure 1 for TOST and ViT (with 12 attention layers, $d = 384$, and $K = 8$); we present more computation metrics for all models in Appendix D.2.

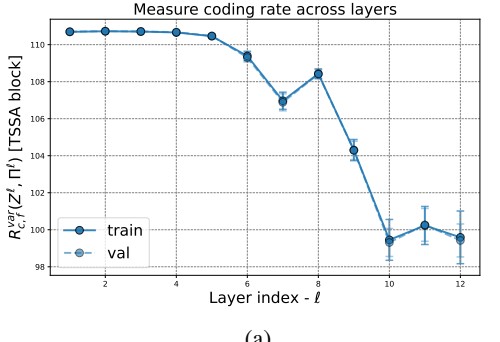
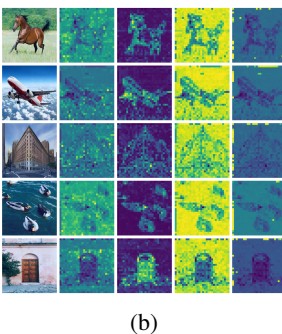

|  |  |
|:---:|:---:|
| (a) | (b) |

Figure 4: (*Left*) **The variational compression term** $R_{c,f}^{\mathrm{var}}(\mathbf{Z}^\ell, \mathbf{\Pi}^\ell)$ **of the `TSSA` outputs $\mathbf{Z}^\ell$ and estimated memberships $\mathbf{\Pi}^\ell$ at different layers $\ell$ of the TOST-S model.** (*Right*) **Visualization of estimated $\mathbf{\Pi}$ for several images.** For an image with $N$ tokens (patches), we visualize each row of the membership matrix $\mathbf{\Pi}$ in the `TSSA` layer by reshaping it into a $\sqrt{N} \times \sqrt{N}$ matrix. Here we visualize membership matrices $\mathbf{\Pi}$ in TOST-S, estimated in layer 9, of each input image.

## 4 EXPERIMENTS

In this section, we conduct experiments to verify and study the properties and performance of our proposed Token Statistics Transformer (TOST) on real-world datasets and tasks. As detailed in Section 3, we design our white-box attention operator by following the principle of MCR$^2$ and unrolling the compression objective defined in equation 4. We thus adopt a simplistic implementation as close to our theoretical derivation as possible. Hence, it is not the goal of this work to show the current implementation of TOST can outperform existing highly engineered architectures. Instead, our empirical studies aim to provide answers and evidence for the following questions:

1. Does our proposed `TSSA` attention operator optimize the compression objective 4 in practice?

2. If we simply replace standard self-attention with our `TSSA` attention operator, leaving the rest of the architecture largely unchanged, do we maintain (or improve) task performance?

We provide positive answers to both questions. First, since our `TSSA` attention operator is mathematically derived via algorithmic unrolling, it should optimize a quantitative objective of the representation, which we show occurs layer-by-layer. Second, we demonstrate that our proposed TOST model, despite its simplicity and efficiency, already performs competitively on real-world large datasets when compared to architectures which use conventional self-attention mechanisms.

**Model architecture.** Since our main focus is the proposed `TSSA` operator as explained in Section 3, we follow a minimalist approach of architecture design by directly replacing attention layers in existing transformer architectures with the `TSSA` operator. Concretely, our realization of TOST inherits most other design choices from XCiT (Ali et al., 2021)[4] and GPT-2 (Radford et al., 2019) for vision and language tasks, respectively. We also test different model sizes by varying the number of attention heads $K$, the token dimension $d$, and number of layers $L$. For supervised classification, we also follow XCiT by inserting a learnable [CLS] token that aggregates features from image patches via a global class attention layer. More details on implementation and model configurations are provided in Appendix C.

**Datasets and optimization.** For vision experiments, we pre-train the proposed TOST models on the ImageNet-1k (Deng et al., 2009) dataset. We also use these pre-trained networks as initialization and fine-tune them on smaller datasets including CIFAR10/100 (Krizhevsky et al., 2009), Oxford Flowers (Nilsback & Zisserman, 2008), Oxford-IIT-Pets (Parkhi et al., 2012) for transfer learning evaluations. We also adopt the Long-Range Arena (Tay et al., 2021) benchmark to analyze the long sequence modeling capability of TOST. For causal language modeling, we train TOST autoregressively on the OpenWebText dataset(Gokaslan et al., 2019) and test the trained model on its test split as well as other datasets as shown in (Radford et al., 2019). Details on training and optimization are provided in Appendix C.3.

---

[4]To align with our theoretical formulation, we remove the Local Patch Interaction (LPI) module from XCiT. See details in Appendix C.2.

Table 2: **Top 1 accuracy of TOST on various datasets with different model sizes when pre-trained on ImageNet.** For ImageNet/ImageNetReaL, we directly evaluate the top-1 accuracy. For other datasets, we pre-train the model on ImageNet and then evaluate the transfer learning performance via fine-tuning.

| Datasets | TOST-T(iny) | TOST-S(mall) | TOST-M(edium) | XCiT-S | XCiT-M | ViT-S | ViT-B(ase) |
|---|---|---|---|---|---|---|---|
| # parameters | 5.8M | 22.6M | 68.1M | 24.9M | 80.2M | 22.1M | 86.6 M |
| ImageNet | 67.3 | 77.9 | 80.3 | 80.5 | 81.5 | 79.8 | 81.8 |
| ImageNet ReaL | 72.2 | 84.1 | 85.6 | 85.6 | 85.9 | 85.6 | 86.7 |
| CIFAR10 | 95.5 | 96.5 | 97.5 | 98.1 | 98.3 | 98.6 | 98.8 |
| CIFAR100 | 78.3 | 82.7 | 84.5 | 86.1 | 87.6 | 88.8 | 89.3 |
| Oxford Flowers-102 | 88.6 | 92.8 | 94.2 | 93.9 | 94.0 | 94.0 | 95.7 |
| Oxford-IIIT-Pets | 85.6 | 91.1 | 92.8 | 92.9 | 94.0 | 92.8 | 94.1 |

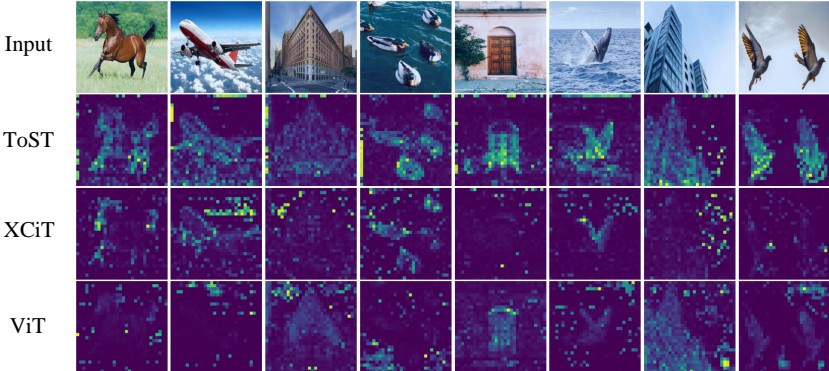

Figure 5: **Comparison of `[CLS]` token attention map visualization.** We take the last head in the penultimate global class attention layer for visualization from TOST-S,XCiT-S, and ViT-S, respectively.

## 4.1 LAYER-WISE ANALYSIS OF THE TSSA OPERATOR

In this sub-section, we aim to answer the first question by providing both qualitative and quantitative results that demonstrate each layer of the proposed TOST attention operator indeed optimizes its designed objective. This layer-wise analysis is possible due to the white-box design of the TSSA operator, similar to studies conducted for CRATE, but different from most black-box models.

**Do TOST attention layers achieve their design goals?** As detailed in Section 3, the TOST attention operator aims to optimize the variational compression term $R_{c,f}^{\mathrm{var}}(\mathbf{Z}, \mathbf{\Pi})$ defined in Equation (4). In order to understand whether the learned attention operator indeed performs such optimization, we explicitly measure and plot the objective value $R_{c,f}^{\mathrm{var}}(\mathbf{Z}^\ell, \mathbf{\Pi}^\ell)$ of the outputs $\mathbf{Z}^\ell$ and group estimations $\mathbf{\Pi}^\ell = \mathbf{\Pi}(\mathbf{Z}^\ell \mid \{\mathbf{U}_k^\ell\}_{k=1}^K)$ of each layer $\ell$. We measure this term on both training and validation set of the ImageNet-1k dataset by taking 500 samples from each. We plot our observations in Figure 4a. The compression term mostly decreases by layer, aligning with the design goal.

**Visualizing membership assignment $\mathbf{\Pi}$ in TOST.** At each layer $\ell$, the estimated membership matrix $\mathbf{\Pi}^\ell$ has the interpretation that each column $\boldsymbol{\pi}_k^\ell \in \mathbb{R}^N$ represents the estimated probabilities for each of the $N$ tokens to be assigned to the $k^{\mathrm{th}}$ subspace. Hence, the $\mathbf{\Pi}$ matrix represents a soft clustering of the $N$ tokens to $K$ clusters, or subspaces, via the $K$ attention heads. In Figure 4b, we visualize the clustering at a selected layer, and observe that foreground image patches are clustered in the same subspace, confirming our interpretation. See Figure 9 for more results.

## 4.2 EVALUATION ON REAL-WORLD VISUAL DATASETS

We now provide evaluations and analysis of TOST models on vision classification tasks, along with comparison against other state-of-the-art vision transformer architectures.

**ImageNet-1k performance and transfer learning results.** In Table 2, we report the top-1 accuracy of TOST on the ImageNet-1k dataset as well as their transfer learning performance on several smaller datasets. We observe that the proposed TOST models achieve comparable ImageNet-1k and transfer learning performance as other empirically engineered architectures, while being significantly more efficient and interpretable with fewer parameters. Notably, we identify promising scaling behavior, where our models consistently improve and the performance gap between TOST and other networks decreases as we scale up the model sizes.

Table 3: **Long-Range Arena (LRA) performance comparison.**

| Model | ListOps | Text | Retrieval | Image | Pathfinder | Avg |
|---|---|---|---|---|---|---|
| Reformer | **37.27** | 56.10 | 53.40 | 38.07 | 68.50 | 50.56 |
| BigBird | 36.05 | 64.02 | 59.29 | 40.83 | 74.87 | 54.17 |
| LinFormer | 16.13 | 65.90 | 53.09 | 42.34 | 75.30 | 50.46 |
| Performer | 18.01 | 65.40 | 53.82 | 42.77 | **77.05** | 51.18 |
| Transformer | 37.11 | 65.21 | 79.14 | 42.94 | 71.83 | 59.24 |
| **ToST (ours)** | 37.25 | **66.75** | **79.46** | **46.62** | 69.41 | **59.90** |
| S4 | 59.60 | 86.82 | 90.90 | 88.65 | 94.20 | 84.03 |

Table 4: **Causal language modeling performance.** (*Left*) We report zero-shot cross entropy loss of GPT2-Base compared with ToST of varying model sizes on the test set of the datasets (lower is better). (*Right*) Relative time and memory usage for base models.

| Model | # params | OWT | Lambada | Wikitext | PTB | Avg ↓ |
|---|---|---|---|---|---|---|
| GPT2-Base | 124M | 2.84 | 4.32 | 4.13 | 5.75 | 4.26 |
| ToST-Base | 110M | 3.20 | 4.98 | 4.77 | 6.39 | 4.84 |
| ToST-Medium | 304M | 2.88 | 4.45 | 4.30 | 5.64 | 4.32 |
| ToST-Large | 655M | 2.72 | 4.32 | 3.99 | 5.03 | 4.02 |

| Model | Length 4k | | Length 8k | |
|---|---|---|---|---|
| | Time | Memory | Time | Memory |
| GPT2-Base | $1\times$ | $1\times$ | $1\times$ | $1\times$ |
| ToST-Base | $0.60\times$ | $0.41\times$ | $0.46\times$ | $0.24\times$ |

**Visualizing attention maps of ToST** We can gain additional understanding of the features learned by ToST based on visualization of the self-attention maps. For the global class attention layers, we extract and visualize the attention map between `[CLS]` token and image patch tokens for each head using the widely adopted approach from (Caron et al., 2021). In Figure 5, we visualize the attention maps in the penultimate layer, for ToST-S, XCiT-S, and ViT-S models. Based on the results, ToST autonomously learns to perform segmentation and clustering without the need for complex self-supervised training techniques or segmentation-related annotations. Compared to the XCiT (Ali et al., 2021) and ViT (Touvron et al., 2021) models, ToST demonstrates more meaningful segmentation results. Figure 10 contains more attention map visualization for each model.

## 4.3 EVALUATION ON LANGUAGE AND LONG SEQUENCE TASKS

**Long sequence modeling.** To assess the performance of ToST on long sequence tasks, we use the Long-Range Arena (LRA) benchmark (Tay et al., 2021) and compare ToST against the transformer and its efficient variants (Kitaev et al., 2020; Zaheer et al., 2020; Wang et al., 2020; Choromanski et al., 2021). The results are reported in Table 3. These results demonstrate promising long-range modeling capability of ToST which outperforms other transformer-based methods on average. We note that this benchmark is currently dominated by state-space models (SSMs) rather than transformer models, but among transformer-based models ToST has highly competitive performance, demonstrating the utility of our `TSSA` attention operator even for long sequence lengths. For completeness, we also include performance results for S4 (Gu et al., 2022), a representative SSM.

**Scalability of ToST for causal language modeling.** We can naturally extend the ToST architecture to perform next-token prediction by introducing a simple causal variant of the `TSSA` operator, as detailed in Appendix B. We present our results in Table 4. We observe that performance of ToST on various text datasets steadily improve as we scale up the size of the model, demonstrating promising scaling behavior. Due to computational constraints, we did not systematically sweep hyperparameters for training ToST for this task and employed the same set as GPT2-Base for training all our models. Still, we find that swapping our `TSSA` attention operator for self-attention largely preserves model performance, despite self-attention largely being heralded as the key component of transformer performance for language modeling tasks. Figure 8 shows that the causal version of ToST is also faster and uses less memory than standard transformers for autoregressive prediction.

## 5 CONCLUSION

In this work, we propose a novel attention operator based on white-box design by unrolled optimization of a novel variational form of the $MCR^2$ objective. Our resulting architecture (ToST) is unique among attention operators in that it does not require computing pairwise interactions between tokens and instead is constructed from a second moment statistic of projected token features. This results in our operator being significantly more efficient than standard attention operators, while still achieving similar performance to comparable transformers. We believe that this work provides an initial demonstration of the tremendous potential in designing novel and efficient deep architectures from mathematical principles.

ACKNOWLEDGMENTS

This work was partially supported by NIH grant R21AI169363, ARO MURI W911NF-17-1-0304, NSF grant 2031985, the Northrop Grumman Mission Systems Research in Applications for Learning Machines (REALM) initiative, and a UC Berkeley College of Engineering Fellowship. Ziyang Wu and Yi Ma acknowledge support from the joint Simons Foundation-NSF DMS grant #2031899, the ONR grant N00014-22-1-2102, the NSF grant #2402951, and Yi Ma also acknowledges partial support from TBSI, InnoHK, and the University of Hong Kong.

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

## A    PROOFS OF THE THEORETICAL RESULTS

Below we present a more refined version of Theorem 1, which easily proves Theorem 1 by taking $p = d$, and also is sufficient to prove Corollary 1.

**Theorem 2.** *Let* $f \colon [0, \infty) \to \mathbb{R}$ *be non-decreasing, concave, and obey* $f(0) = 0$, *and let* $F \colon \mathrm{PSD}(d) \to \mathbb{R}$ *have the form*

$$F(\mathbf{M}) = \sum_{i=1}^{d} f(\lambda_i(\mathbf{M})), \qquad \forall \mathbf{M} \in \mathrm{PSD}(d). \tag{12}$$

*Then, for all* $\mathbf{M} \in \mathrm{PSD}(d)$ *and all* $\mathbf{Q} \in \mathrm{O}(d, p)$ *with* $p \leq d$ *such that* $\mathrm{image}(\mathbf{M}) \subseteq \mathrm{image}(\mathbf{Q})$, *we have*

$$F(\mathbf{M}) \leq \sum_{i=1}^{p} f((\mathbf{Q}^{\top}\mathbf{M}\mathbf{Q})_{ii}). \tag{13}$$

*The inequality in equation 13 is achieved with equality for any* $\mathbf{Q}$ *which diagonalizes* $\mathbf{M}$, *i.e.,* $\mathbf{Q}^{\top}\mathbf{M}\mathbf{Q}$ *is diagonal. Moreover, if* $f$ *is strictly concave, then the inequality in equation 13 is achieved with equality if and only if* $\mathbf{Q}$ *diagonalizes* $\mathbf{M}$.

*Proof.* First, we have that $\mathrm{rank}(\mathbf{M}) \leq \mathrm{rank}(\mathbf{Q}) = p$. Thus, let $\mathbf{M} = \mathbf{U}\boldsymbol{\Lambda}\mathbf{U}^{\top}$ be a spectral decomposition of $\mathbf{M}$, where $\boldsymbol{\Lambda} \in \mathbb{R}^{d \times d}$ is diagonal with non-negative entries $\lambda_i(\mathbf{M})$ for $i \in [d]$, and $\mathbf{U} \in \mathrm{O}(d)$. In particular, we can write

$$\boldsymbol{\Lambda} = \begin{bmatrix} \boldsymbol{\Lambda}_p & \mathbf{0} \\ \mathbf{0} & \mathbf{0} \end{bmatrix}, \qquad \mathbf{U} = \begin{bmatrix} \mathbf{U}_p & \tilde{\mathbf{U}} \end{bmatrix} \tag{14}$$

where $\boldsymbol{\Lambda}_p \in \mathbb{R}^{p \times p}$ is a diagonal matrix with non-negative entries $\lambda_i(\mathbf{M})$ for $i \in [p]$, $\mathbf{U}_p \in \mathrm{O}(d, p)$ and $\tilde{\mathbf{U}}_p \in \mathrm{O}(d, d - p)$, such that $\mathrm{image}(\mathbf{Q}) = \mathrm{image}(\mathbf{U}_p)$, so that $\mathrm{image}(\mathbf{M}) \subseteq \mathrm{image}(\mathbf{U}_p)$ and $\mathrm{image}(\mathbf{Q})$ is orthogonal (in the sense of linear subspaces) to $\mathrm{image}(\tilde{\mathbf{U}})$.

We begin by noting the simple fact that if $\mathbf{Q}$ diagonalizes $\mathbf{M}$, then we have $\mathbf{Q}^{\top}\mathbf{M}\mathbf{Q} = \mathbf{P}^{\top}\boldsymbol{\Lambda}_p\mathbf{P}$ for some permutation matrix $\mathbf{P} \in \mathrm{O}(p)$. This gives

$$\sum_{i=1}^{p} f((\mathbf{Q}^{\top}\mathbf{M}\mathbf{Q})_{ii}) = \sum_{i=1}^{p} f((\mathbf{P}^{\top}\boldsymbol{\Lambda}_p\mathbf{P})_{ii}) = \sum_{i=1}^{p} f((\boldsymbol{\Lambda}_p)_{ii}) = \sum_{i=1}^{p} f(\lambda_i(\mathbf{M})) \tag{15}$$

$$= \sum_{i=1}^{d} f(\lambda_i(\mathbf{M})) = F(\mathbf{M}), \tag{16}$$

showing that the inequality equation 13 is met with equality for a $\mathbf{Q}$ which diagonalizes $\mathbf{M}$.

We now show that the inequality holds for a general $\mathbf{Q} \in \mathrm{O}(d, p)$ such that $\mathrm{image}(\mathbf{M}) \subseteq \mathrm{image}(\mathbf{Q})$. Define $\mathbf{W} = \mathbf{U}^{\top}\mathbf{Q} \in \mathbb{R}^{d \times p}$. Then we have

$$\mathbf{W} = \mathbf{U}^{\top}\mathbf{Q} = \begin{bmatrix} \mathbf{U}_p^{\top} \\ \tilde{\mathbf{U}}^{\top} \end{bmatrix} \mathbf{Q} = \begin{bmatrix} \mathbf{U}_p^{\top}\mathbf{Q} \\ \tilde{\mathbf{U}}^{\top}\mathbf{Q} \end{bmatrix} = \begin{bmatrix} \mathbf{U}_p^{\top}\mathbf{Q} \\ \mathbf{0} \end{bmatrix}. \tag{17}$$

Now we have

$$\mathbf{W}^{\top}\mathbf{W} = \mathbf{Q}^{\top}\mathbf{U}\mathbf{U}^{\top}\mathbf{Q} = \mathbf{Q}^{\top}\mathbf{I}\mathbf{Q} = \mathbf{Q}^{\top}\mathbf{Q} = \mathbf{I}, \tag{18}$$

so that $\mathbf{W}$ has orthonormal columns, i.e., $\mathbf{W} \in \mathrm{O}(d, p)$. In particular, writing this in a different way,

$$\mathbf{I} = \mathbf{W}^{\top}\mathbf{W} = \begin{bmatrix} \mathbf{Q}^{\top}\mathbf{U}_p & \mathbf{0} \end{bmatrix} \begin{bmatrix} \mathbf{U}_p^{\top}\mathbf{Q} \\ \mathbf{0} \end{bmatrix} = \mathbf{Q}^{\top}\mathbf{U}_p\mathbf{U}_p^{\top}\mathbf{Q} = (\mathbf{U}_p^{\top}\mathbf{Q})^{\top}(\mathbf{U}_p^{\top}\mathbf{Q}), \tag{19}$$

so that $\mathbf{U}_p^{\top}\mathbf{Q} \in \mathrm{O}(p)$.

Now note that we have

$$(\mathbf{Q}^{\top}\mathbf{M}\mathbf{Q})_{ii} = ((\mathbf{U}\mathbf{W})^{\top}\mathbf{M}(\mathbf{U}\mathbf{W}))_{ii} = (\mathbf{W}^{\top}\mathbf{U}^{\top}\mathbf{M}\mathbf{U}\mathbf{W})_{ii} = (\mathbf{W}^{\top}\boldsymbol{\Lambda}\mathbf{W})_{ii} \tag{20}$$

$$= \sum_{j=1}^{d} \lambda_j(\mathbf{M})\mathbf{W}_{ji}^2. \tag{21}$$

Since $\mathbf{W}$ has orthonormal columns, $\sum_{j=1}^{d} \mathbf{W}_{ji}^2 = 1$. Thus $(\mathbf{Q}^\top \mathbf{M} \mathbf{Q})_{ii}$ is a convex combination of the eigenvalues of $\mathbf{M}$, including some zeros. Using the concavity of $f$, this yields

$$\sum_{i=1}^{p} f((\mathbf{Q}^\top \mathbf{M} \mathbf{Q})_{ii}) = \sum_{i=1}^{p} f\left(\sum_{j=1}^{d} \lambda_j(\mathbf{M}) \mathbf{W}_{ji}^2\right) \tag{22}$$

$$\geq \sum_{i=1}^{p} \sum_{j=1}^{d} \mathbf{W}_{ji}^2 f(\lambda_j(\mathbf{M})) \tag{23}$$

$$= \sum_{j=1}^{d} f(\lambda_j(\mathbf{M})) \sum_{i=1}^{p} \mathbf{W}_{ji}^2 = \sum_{j=1}^{p} f(\lambda_j(\mathbf{M})) \sum_{i=1}^{p} \mathbf{W}_{ji}^2 \tag{24}$$

$$= \sum_{j=1}^{p} f(\lambda_j(\mathbf{M})) \sum_{i=1}^{p} (\mathbf{U}_p^\top \mathbf{Q})_{ji}^2 \tag{25}$$

$$= \sum_{j=1}^{p} f(\lambda_j(\mathbf{M})) = \sum_{j=1}^{d} f(\lambda_j(\mathbf{M})) = F(\mathbf{M}). \tag{26}$$

In particular, $\sum_{i=1}^{p} (\mathbf{U}_p^\top \mathbf{Q})_{ji}^2 = 1$ because $\mathbf{U}_p^\top \mathbf{Q}$ is an orthogonal matrix, thus having unit-norm rows and columns. This proves the inequality equation 13.

Regarding the equality case, we note that the only inequality used to derive equation 13 is the concavity condition on $f$. If $f$ is strictly concave, then this inequality is met with equality if and only if, for each $i$, there exists $j^\star$ such that $\sum_{j=1}^{d} \lambda_j(\mathbf{M}) \mathbf{W}_{ji}^2 = \lambda_{j^\star}(\mathbf{M})$. Namely, the inequality is met with equality if and only if for each $i$, if $\mathbf{W}_{j_1 i} \neq 0$ and $\mathbf{W}_{j_2 i} \neq 0$ then $\lambda_{j_1}(\mathbf{M}) = \lambda_{j_2}(\mathbf{M})$. For example, if $\mathbf{M}$ has distinct eigenvalues then $\mathbf{W}$ is a truncated permutation matrix. In general, however, a $\mathbf{W}$ with these properties implies that $\mathbf{Q} = \mathbf{U}\mathbf{W}$ diagonalizes $\mathbf{M}$. This shows that if $f$ is strictly concave then the inequality in equation 13 is met with equality if and only if $\mathbf{Q}$ diagonalizes $\mathbf{M}$. $\qquad\square$

# B    EXTENSION TO CAUSAL ToST ATTENTION

To enable auto-regressive training / inference similar to GPT, the ToST attention mechanism illustrated in the main text needs to be adapted to a *causal* version. Namely, for the $t$-th token, it can only attend to tokens from earlier timesteps $\{1, \ldots, t-1\}$.

In conventional self-attention mechanism, such operation can be achieved by applying a causal mask on the attention matrix $A \in \mathbb{R}^{T \times T}$, where $T$ is the total number of tokens. The causal mask $M$ is a $T$-by-$T$ lower-triangular matrix of binary values (i.e. 0 or 1). The resulting causal attention matrix $M \odot A$ is therefore also lower-triangular and prevents any token from attending to future tokens.

In ToST attention, such operation however requires some tweaking. Let us first revisit the formulation of the ToST attention mechanism:

$$\mathbf{z}_j^+ = \mathbf{z}_j - \frac{\tau}{n} \sum_{k=1}^{K} \Pi_{j,k} \mathbf{U}_k \mathrm{Diag}\left(\nabla f\left[(\mathbf{U}_k^\top \mathbf{Z})^{\odot 2} \frac{\boldsymbol{\pi}_k}{\langle \boldsymbol{\pi}_k, \mathbf{1}\rangle}\right]\right) \mathbf{U}_k^\top \mathbf{z}_j, \quad \forall j \in [n]. \tag{27}$$

To enforce causality between tokens, the key is to make sure for generating token $\mathbf{z}_j^+$, we should only rely on statistics computed from tokens $\{\mathbf{z}_1, \ldots, \mathbf{z}_j\}$. Assume we have already computed $\mathbf{\Pi}$[5], the causal version of ToST becomes:

$$\mathbf{z}_j^+ = \mathbf{z}_j - \frac{\tau}{j} \sum_{k=1}^{K} \Pi_{j,k} \mathbf{U}_k \mathrm{Diag}\left(\nabla f\left[(\mathbf{U}_k^\top \mathbf{Z})^{\odot 2} \cdot \frac{\mathbf{I}_{j,n} \boldsymbol{\pi}_k}{\langle \mathbf{I}_{j,n} \boldsymbol{\pi}_k, \mathbf{1}\rangle}\right]\right) \mathbf{U}_k^\top \mathbf{z}_j, \quad \forall j \in [n]. \tag{28}$$

---

[5]Note $\mathbf{\Pi}$ itself may be causal depending on how it is computed. For now, just assume we have computed $\mathbf{\Pi}$ and we will explain how to handle $\mathbf{\Pi}$ with a more complex initialization scheme in the next section.

where we define

$$\mathbf{I}_{j,n} = \mathrm{Diag}(\underbrace{1,\ldots,1}_{j}, \underbrace{0,\ldots,0}_{n-j}). \tag{29}$$

Notice here we have effectively restricted the attention operator to only use the first $j$ tokens, preventing influence from future tokens. This mechanism, named Causal-ToST, could be efficiently implemented in practice with a cumulative sum (i.e. `cumsum`) operation. This is an intuitive way to encode causality into our attention mechanism, in contrast to the XCA opeartor in XCiT, which does not have a direct way of implementing causal attention. We will introduce more technical details on the implementation of this causal variant in the next section.

## B.1 CONNECTION TO OTHER EFFICIENT ARCHITECTURES

It is worth noting that there exist other works on efficient alternatives to self-attention. In particular, recent works on deep state-space models (Gu et al., 2022) or variants of linear attention (Yang et al., 2025) have become increasingly popular and similarly enjoy linear computational complexity. In this subsection, we address connections and distinctions of our work to these methods. First, SSMs and linear attention variants turn out to be deeply related and they can be viewed under a unified framework of *test-time regression* (Wang et al., 2025). From this perspective, these methods can be similarly viewed as algorithmic unrolling: these operators are obtained via optimization (e.g. gradient descent) from some learning objectives. The fundamental differences between ToST and these efficient alternatives thus primarily lie in the modeling assumption and objective functions. SSMs, for example, are primarily derived from the state-space representation modeled by some first-order (continuous) differential equations. This choice entails distinct design choices for SSM architectures such as discretization and efficient convolution operators. On the other hand, ToST is derived from a compression-based objective on the space of discrete tokens. Similar to self-attention, our resulting operator acts on tokens by aggregating token-level information (but without direct pairwise interaction). Hence, our work doesn't share many design choices of SSMs and is placed much closer to transformers. Consequently, we conduct empirical studies by directly swapping the attention module of standard transformers with our derived ToST attention. Nevertheless, we believe it is an interesting direction to explore how related these two formulations are, but this is beyond the scope of this paper and we leave a deeper exploration of their connections to future work.

## C EXPERIMENT DETAILS

Here we present additional settings and results for our implementation and experiments.

## C.1 IMPLEMENTATION DETAILS

For ease of optimization and implementation, below we make a few modifications to the conceptual architecture presented in Section 3.

**Overparameterization in `TSSA`.** First notice we can express the TSSA term in Equation (11) as:

$$\mathrm{TSSA}(\mathbf{Z} \mid \{\mathbf{U}_k\}_{k=1}^{K}) = -\frac{\tau}{n} \left[\mathbf{U}_1, \ldots, \mathbf{U}_K\right] \begin{bmatrix} \mathbf{D}(\mathbf{Z}, \boldsymbol{\pi}_1 \mid \mathbf{U}_1)\mathbf{U}_1^\top \mathbf{Z}\mathrm{Diag}(\boldsymbol{\pi}_1) \\ \vdots \\ \mathbf{D}(\mathbf{Z}, \boldsymbol{\pi}_K \mid \mathbf{U}_K)\mathbf{U}_K^\top \mathbf{Z}\mathrm{Diag}(\boldsymbol{\pi}_K) \end{bmatrix}, \tag{30}$$

where $\mathbf{D}$ is defined in equation 6. Following Yu et al. (2024), we replace the term $(\tau/n)\left[\mathbf{U}_1, \ldots, \mathbf{U}_K\right]$ with a learnable matrix $\mathbf{W} \in \mathbb{R}^{d \times pK}$. Expanding the definition of $\mathbf{D}$, the TSSA operator becomes:

$$\mathrm{TSSA}(\mathbf{Z} \mid \{\mathbf{U}_k\}_{k=1}^{K}) = -\mathbf{W} \begin{bmatrix} \mathrm{Diag}(\nabla f[(\mathbf{U}_1^\top \mathbf{Z})^{\odot 2}\boldsymbol{\pi}_1/\langle \boldsymbol{\pi}_1, \mathbf{1}\rangle])\mathbf{U}_1^\top \mathbf{Z}\mathrm{Diag}(\boldsymbol{\pi}_1) \\ \vdots \\ \mathrm{Diag}(\nabla f[(\mathbf{U}_K^\top \mathbf{Z})^{\odot 2}\boldsymbol{\pi}_K/\langle \boldsymbol{\pi}_K, \mathbf{1}\rangle])\mathbf{U}_K^\top \mathbf{Z}\mathrm{Diag}(\boldsymbol{\pi}_K) \end{bmatrix}. \tag{31}$$

**Absorbing constant coefficients.** We absorb the coefficients $d/\epsilon^2$ in $\nabla f(x) = (d/\epsilon^2)(1 + (d/\epsilon^2)x)^{-1}$ in Equation (11) into the trainable parameter $\{\mathbf{U}_k\}_{k=1}^{K}$. Therefore, the non-linear activation function becomes $\nabla f(x) = (1 + x)^{-1}$.

**Estimating membership $\mathbf{\Pi}$.** When estimating $\mathbf{\Pi}$ as in Equation (10), we perform $\ell_2$ normalization on the projected tokens $\mathbf{U}_k^\top \mathbf{Z} \in \mathbb{R}^{p \times n}$ along each feature dimension, for each head. That is, we restrict the norm of each row of the projected tokens to be 1. This normalization scheme is *only* applied to tokens in the computation of $\mathbf{\Pi}$ and does not impact anything else in Equation (11). We found this design to substantially improve and stabilize the training of ToST models. To express the aforementioned normalization, denote the $i$-th row of $\mathbf{U}_k^\top \mathbf{Z}$ as $(\mathbf{U}_k^\top \mathbf{Z})_{i,:}$. Then define

$$\mathbf{y}_k = \begin{bmatrix} 1/\|(\mathbf{U}_k^\top \mathbf{Z})_{1,:}\|_2 \\ \vdots \\ 1/\|(\mathbf{U}_k^\top \mathbf{Z})_{p,:}\|_2 \end{bmatrix}, \qquad \mathbf{u}_k^j \doteq \mathbf{U}_k^\top \mathbf{z}_j \odot \mathbf{y}_k. \tag{32}$$

Then,

$$\mathbf{\Pi} = \begin{bmatrix} \boldsymbol{\nu}(\mathbf{z}_1 \mid \{\mathbf{U}_k\}_{k=1}^K)^\top \\ \vdots \\ \boldsymbol{\nu}(\mathbf{z}_n \mid \{\mathbf{U}_k\}_{k=1}^K)^\top \end{bmatrix}, \tag{33}$$

$$\text{where} \quad \boldsymbol{\nu}(\mathbf{z}_j \mid \{\mathbf{U}_k\}_{k=1}^K) \doteq \text{softmax}\left(\frac{1}{2\eta}\begin{bmatrix} \|\mathbf{u}_1^j\|_2^2 \\ \vdots \\ \|\mathbf{u}_K^j\|_2^2 \end{bmatrix}\right), \quad \forall j \in [n]. \tag{34}$$

In the causal variant of $\texttt{TSSA}$ operator as previously introduced (Appendix B), this normalization scheme means that, each token has its own corresponding group assignment matrix $\mathbf{\Pi}$ that only relies on earlier tokens for normalization. Denote $\mathbf{\Pi}^j \in \mathbb{R}^{p \times j}$ a group assignment matrix corresponding to the $j$-th token. Define

$$\mathbf{y}_k^j = \begin{bmatrix} 1/\|(\mathbf{I}_{j,n} \cdot \mathbf{U}_k^\top \mathbf{Z}^\ell)_{1,:}\|_2 \\ \vdots \\ 1/\|(\mathbf{I}_{j,n} \cdot \mathbf{U}_k^\top \mathbf{Z}^\ell)_{p,:}\|_2 \end{bmatrix}, \qquad \text{where (recall)} \qquad \mathbf{I}_{j,n} = \text{Diag}(\underbrace{1,\ldots,1}_{j},\underbrace{0,\ldots,0}_{n-j}). \tag{35}$$

Then,

$$\mathbf{\Pi}^j = \begin{bmatrix} \boldsymbol{\nu}^j(\mathbf{z}_1 \mid \{\mathbf{U}_k\}_{k=1}^K)^\top \\ \vdots \\ \boldsymbol{\nu}^j(\mathbf{z}_j \mid \{\mathbf{U}_k\}_{k=1}^K)^\top \end{bmatrix}, \tag{36}$$

$$\text{where} \quad \boldsymbol{\nu}^j(\mathbf{z}_i \mid \{\mathbf{U}_k\}_{k=1}^K) \doteq \text{softmax}\left(\frac{1}{2\eta}\begin{bmatrix} \|\mathbf{U}_1^\top \mathbf{z}_i \odot \mathbf{y}_1^j\|_2^2 \\ \vdots \\ \|\mathbf{U}_K^\top \mathbf{z}_i \odot \mathbf{y}_K^j\|_2^2 \end{bmatrix}\right), \quad \forall j \in [n]. \tag{37}$$

Similar to our analysis in Appendix B, this normalization scheme can be implemented using the $\texttt{cumsum}$ operation. However, one can notice that a naive implementation would require an entirely different $\mathbf{\Pi}^j$ to be computed every time a new token arrives at time step $t$, as each column $k \leq t - 1$ corresponding to earlier time steps needs to be recomputed. This would lead to quadratic computational and memory complexity. In practice, instead of recomputing earlier columns, we find that using a zero-initialized learnable additive bias along both the head and the sequence dimensions before the softmax operator is sufficient. Denote the bias $\mathbf{b}_k^j$ for $k$-th head and $j$-th token. Then,

$$\mathbf{\Pi} = \begin{bmatrix} \boldsymbol{\nu}(\mathbf{z}_1 \mid \{\mathbf{U}_k\}_{k=1}^K)^\top \\ \vdots \\ \boldsymbol{\nu}(\mathbf{z}_n \mid \{\mathbf{U}_k\}_{k=1}^K)^\top \end{bmatrix}, \tag{38}$$

$$\text{where} \quad \boldsymbol{\nu}(\mathbf{z}_j \mid \{\mathbf{U}_k\}_{k=1}^K) \doteq \text{softmax}\left(\frac{1}{2\eta}\begin{bmatrix} \|\mathbf{U}_1^\top \mathbf{z}_j \odot \mathbf{y}_1^j\|_2^2 + \mathbf{b}_1^j \\ \vdots \\ \|\mathbf{U}_K^\top \mathbf{z}_j \odot \mathbf{y}_K^j\|_2^2 + \mathbf{b}_K^j \end{bmatrix}\right), \quad \forall j \in [n]. \tag{39}$$

Therefore, we still have Equation (28) as the causal variant of ToST.

We provide a PyTorch-style pseudocode in Appendix E that reflects the architecture modifications introduced above. In particular, Algorithm 1 and Algorithm 2 implement the $\texttt{TSSA}$ attention layer and its causal variant, respectively.

Table 5: **Configurations for the transformer models in image classification experiments.**

|  | ToST-T | ToST-S | ToST-M | XCiT-S | XCiT-M | ViT-S | ViT-B |
|---|---|---|---|---|---|---|---|
| # parameters | 5.8M | 22.6M | 68.1M | 24.9M | 80.2M | 22.1M | 86.6 M |
| # attention heads $K$ | 4 | 8 | 8 | 8 | 8 | 6 | 12 |
| # layers $L$ | 12 | 12 | 24 | 12 | 24 | 12 | 12 |
| # feature dimension $d$ | 192 | 384 | 512 | 384 | 512 | 384 | 768 |
| # head dimension $p$ | 48 | 48 | 64 | 48 | 64 | 64 | 64 |

Table 6: **Configurations for ToST models in language modeling experiments.**

|  | ToST-Base | ToST-Medium | ToST-Large |
|---|---|---|---|
| # parameters | 110M | 304M | 655M |
| # attention heads $K$ | 12 | 16 | 20 |
| # layers $L$ | 12 | 24 | 36 |
| # feature dimension $d$ | 768 | 1024 | 1280 |
| # head dimension $p$ | 64 | 64 | 64 |

## C.2    MODEL CONFIGURATIONS

In the visual classification experiments conducted in Section 4, we focus on three types of models: the proposed ToST, XCiT, and ViT models. As detailed in the main text, our implementation of ToST inherits choices of other non-attention layers from XCiT. One important modification we make is the removal of Local Patch Interaction (LPI) module, which is designed to encourage explicit between-token interactions and does not align with our theoretical derivation. Therefore, we prevent any form of explicit interactions among tokens by removing the LPI block. We remove this module for XCiT as well for fair comparison. Hence, we train all ToST and XCiT models reported in this paper. One notable design choice of XCiT is the addition of two class attention layers after the last XCiT block. The goal of these layers is to aggregate features from the processed patch tokens and then write to a learnable [CLS] token via conventional self-attention. These layers perform attention only between the [CLS] token and patch tokens, thus incurring $\mathcal{O}(n)$ computational complexity. For ViT models, we directly use the checkpoints and results reported in Touvron et al. (2021). Specifically, we use the ViT-S(mall) and ViT-B(ase) models, as they contain parameters of similar size as XCiT-S(mall) and XCiT-M(edium), respectively. To summarize, we present details about the configurations of these models in Table 5.

In the causal language modeling experiments, we train the causal variant of ToST of three different sizes and closely follow GPT-2 (Radford et al., 2019) family of models. We present their configurations in Table 6. Please see Appendix C.3 for other training details. One thing to note is that we adopt the same hyperparameters as in (Karpathy, 2022) for all sizes due to the limitation of computing resources. A more careful tuning certainly could lead to better performance for ToST.

## C.3    TRAINING SETUP

**Pre-training on ImageNet-1k.** We train our models using the AdamW optimizer with a learning rate of $2 \times 10^{-4}$ for 400 epochs throughout our pre-training experiments. We configure our batch size to be 2048 for all our training experiments. All images are reshaped into resolutions of $224 \times 224$ and we tokenize each patch of size $16 \times 16$ into patch embeddings. For the other hyperparameters including data augmentation strategies, we adopt the *exact same* recipe as in (Ali et al., 2021). Please refer to this prior work for details. We conduct all pre-training experiments on 128 NVIDIA V100 GPUs.

**Fine-tuning.** We conduct transfer learning experiments by using the pretrained ToST, XCiT, and ViT models as initialization and fine-tuning them on the following target datasets: CIFAR10/100, Oxford Flowers-102, Oxford-IIIT-Pets. We use batch size of 256 throughout our fine-tuning experiments. For other settings, we adopt the same hyperparameter setup and pipeline as in (Yu et al., 2024). Regarding computational resources, we conducted experiments on two NVIDIA RTX 4090 GPUs.

Table 7: **Configurations for our ToST models in experiments on the Long Range Arena benchmark.**

| Task | # layers $L$ | feature dim. $d$ | # heads $K$ | MLP dim. | pre-norm | batch size | learning rate | dropout | weight decay | total iter. | warmup iter. | learning rate decay |
|---|---|---|---|---|---|---|---|---|---|---|---|---|
| ListOps | 6 | 512 | 8 | 2048 | True | 64 | 1e-4 | 0.1 | 1e-2 | 90000 | 3000 | inverse sqrt. |
| Text | 4 | 256 | 4 | 1024 | True | 32 | 1e-4 | 0.1 | 1e-2 | 25000 | 10000 | linear |
| Retrieval | 6 | 128 | 4 | 512 | True | 64 | 1e-4 | 0.1 | 4e-2 | 91960 | 10000 | linear |
| Image | 8 | 64 | 8 | 128 | True | 64 | 1e-4 | 0.0 | 2e-2 | 180000 | 9000 | linear |
| Pathfinder | 2 | 64 | 2 | 128 | False | 256 | 2e-4 | 0.1 | 0 | 200000 | 312 | linear |

Table 8: **Model ablation.** Test accuracy on CIFAR10 after 400 epochs for a ToST-S base model with various combinations of $\ell_2$ normalization and non-attention blocks.

| Normalization | Non-Attention Block | Acc. |
|---|---|---|
| Yes | MLP | 91.2 |
| Yes | ISTA | 87.4 |
| No | MLP | 85.2 |
| No | ISTA | 80.6 |

**Long-Range Arena (LRA) Benchmark.** We implement ToST on top of an off-the-shelf PyTorch version[6] of the LRA benchmark and assess ToST on tasks of long list operations (ListOps; Nangia & Bowman, 2018) , byte-level text classification (Text; Maas et al., 2011), document retrieval (Retrieval; Radev et al., 2013), image classification (Image; Krizhevsky et al., 2009), Pathfinder (Linsley et al., 2018), and its extremely long version (Path-X; Tay et al. 2021). Due to the reported issues[7] with the Pathfinder task, we use a different Pytorch implementation[8] of LRA to evaluate ToST on the task of Pathfinder. The hyperparameters of ToST are summarized in Table 7.

**Causal Language Modeling.** We autoregressively train ToST models of sizes corresponding to GPT2-Base, Medium and Large following Radford et al. (2019), by substituting the self-attention operator with our proposed causal version of TOST, with the same pre-processing and post-processing steps, on the OpenWebText dataset (Gokaslan et al., 2019). Our implementation is based on Karpathy (2022). We use a context length of 1024, and optimize the models using AdamW optimizer (Loshchilov & Hutter, 2019) for 600,000 iterations, with batch size 480. We set the learning rate to 0.0006 with 2000 warm up iterations and cosine decay, weight decay to 0.1. For more details, we refer readers to Radford et al. (2019); Karpathy (2022).

# D ADDITIONAL EXPERIMENTAL RESULTS

## D.1 ABLATION STUDIES ON MODEL DESIGN CHOICES.

Recall that we have made a few design choices for our implementation of ToST as detailed in Appendix C.1. Additionally, differently from CRATE (Yu et al., 2024), we use a generic two-layer MLP instead of the sparsity-promoting ISTA block in our implementation. In particular, we find the following two components to have a non-trivial impact on the performance of our proposed model:

1. $\ell_2$ normalization when computing the membership matrix $\boldsymbol{\Pi}$;

2. MLP (as adopted in ViT, XCiT, etc) instead of ISTA (as adopted in CRATE).

We study how these choices affect our model performance and present our observations in Table 8. Specifically, we vary these design choices and train each model variant on CIFAR10 for 400 epochs. Our results demonstrate the necessity of each component in our model design.

---

[6] https://github.com/facebookresearch/mega
[7] https://github.com/facebookresearch/mega/issues/3
[8] https://github.com/pkuzengqi/Skyformer

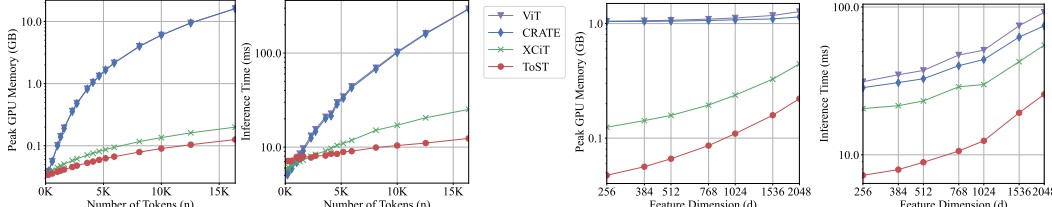

Figure 6: **Memory usage and inference time of attention operators of different transformer architectures: ViT (Dosovitskiy et al., 2020), CRATE (Yu et al., 2023), XCiT (Ali et al., 2021), and ToST (ours).** (*Left*) Each model has $K = 8$ heads and $d = 384$ feature dimensions at every layer with varying number $n$ of input tokens. (*Right*) Each model has $n = 4096$ tokens and $K = 8$ heads at every layer with varying feature dimension $d$. The y-axes are in log-scale.

### D.2 ADDITIONAL COMPARISON OF MEMORY AND TIME COMPLEXITY OF ATTENTION OPERATORS

Beyond the theoretical comparison of the memory and time complexity of attention operators of different architectures in Table 1, we further conduct experiments to provide empirical evidence by evaluating real world memory cost and inference speed on an NVIDIA H800 GPU. Specifically, we measure inference time and memory footprint for 12 attention layers for each of the tested architectures with batch size of 1. For each configuration we take 1 warm-up forward pass and then run 1000 forward passes to record the average inference time.

**Smaller models with $K = 8$ heads.** Figure 6 reports the memory and inference time of attention operators of different transformers, where the y-axes are in log scale for better readability. The two subplots on the left are the setting shown in Figure 1, where we fix feature dimension $d = 384$ and vary the number $n$ of tokens, while for the two subplots on the right, we fix $n = 4096$ tokens and vary the feature dimension $d$. Notably, in all cases, our ToST's attention operator has a much lower computational and memory footprint than the attention of other transformers, which includes the common ViT transformers as well as the reduced-complexity XCiT transformers. Indeed, for $10k$ tokens, Figure 6 (Left) shows that ToST is nearly 10 times faster than ViT for inference and uses roughly 100 times less GPU memory. This aligns well with our theoretical comparison in Table 1.

**Larger models with $K = 32$ heads.** We now move to larger models, where we fix $K = 32$ heads and $d = 4096$ feature dimensions, and vary the number $n$ of tokens. Figure 7 presents the memory and inference time of attention operators of different transformers, where again the y-axes are in log scale. It can be seen that the same trend (as in the smaller models case) shows up, and ToST uses lower memory and inference time than ViT, CRATE, and XCiT.

**Causal-ToST vs GPT.** Here we compare the memory and time complexity of different architectures on GPT-like auto-regressive generation (i.e. next token prediction) tasks. Following the config of GPT2-base, we fix $K = 12$ heads and $d = 768$ feature dimensions, and vary the number $n$ of tokens for both architectures. Figure 8 presents the memory and inference time of Causal-ToST and GPT-2. Similarly, ToST uses lower memory and inference time than GPT-2.

### D.3 ADDITIONAL VISUALIZATION OF MEMBERSHIP MATRIX AND ATTENTION MAPS

We here visualize the membership matrix $\Pi$ learned in ToST-S in different layers in Figure 9, as well as the `[CLS]` attention maps from ToST-S, XCiT-S (Ali et al., 2021), and ViT-S (Touvron et al., 2021) in Figure 10 for better interpretation of the semantic clustering property of ToST, as discussed in Section 4.1.

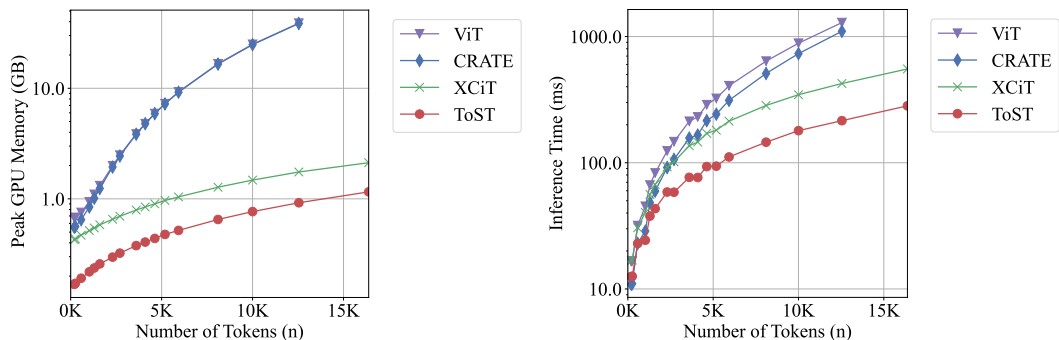

Figure 7: **Memory usage and inference time of attention operators of different transformer architectures with varying numbers** $n$ **of input tokens.** Each model has $K = 32$ heads and $d = 4096$ feature dimension at every layer. The y-axes are in log-scale.

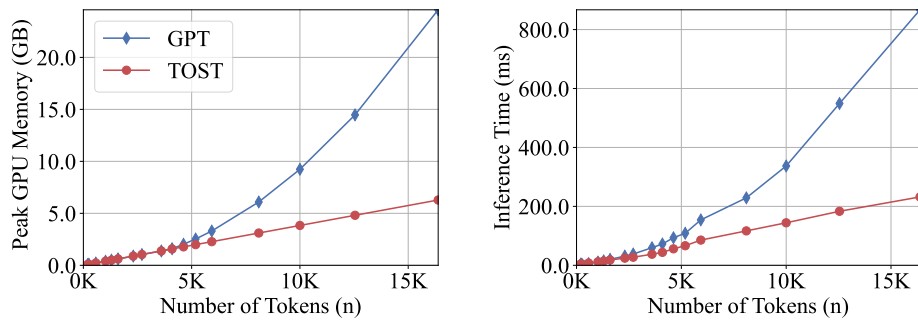

Figure 8: Causal version of our ToST architecture **is also faster and uses less memory** than standard transformer architectures (e.g. GPT-2) on auto-regressive generation tasks. We adopt the Base model size for both architectures.

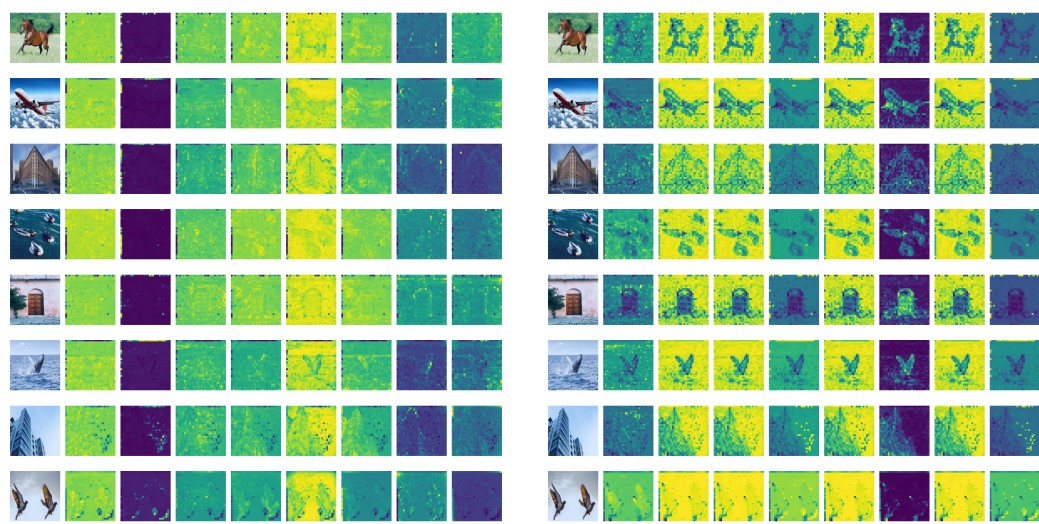

Figure 9: **Visualization of membership matrices $\Pi$ from ToST-S, estimated in layer 5 (*left*) and 9 (*right*) of each input image.**

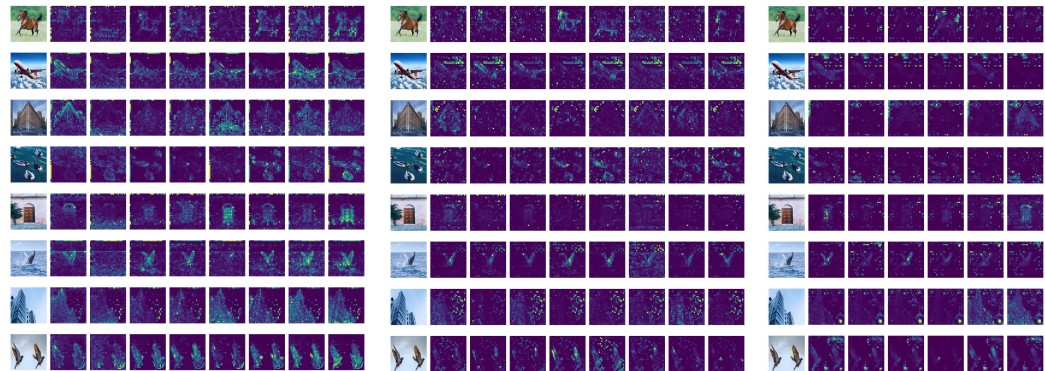

Figure 10: **Comparison of `[CLS]` token attention map visualization in the penultimate layer from ToST-S (*left*), XCiT-S (Ali et al., 2021) (*middle*), and ViT-S (Touvron et al., 2021) (*right*).** Each row is a different image and each column is a different head. Note that the ViT-S model has $K = 6$ heads instead of $K = 8$.

### D.4 VISUALIZING LAYER-WISE SUBSPACES IN TSSA:

As detailed in Section 3.3, we do not, in practice, strictly enforce the orthogonality of the $\mathbf{U}$ columns or require the direct sum of the subspaces defined by $\mathbf{U}_{[K]}$ to cover the entire $\mathbb{R}^p$ space. Instead, we make these projection matrices learnable via backprop for some target tasks (e.g. image classification). We now visualize the learned $\mathbf{U}^{\ell}_{[K]}$ matrices of different layers in the TSSA block of a trained ToST-S model on ImageNet-1k. In Figure 11, we first normalize the columns in each $\mathbf{U}^{\ell}_{k}$, then we visualize $[\mathbf{U}^{\ell}_1, \ldots, \mathbf{U}^{\ell}_K]^{\top}[\mathbf{U}^{\ell}_1, \ldots, \mathbf{U}^{\ell}_K] \in \mathbb{R}^{pK \times pK}$. The $(i, j)^{\text{th}}$ block in each sub-figure corresponds to $(\mathbf{U}^{\ell}_i)^{\top}\mathbf{U}^{\ell}_j$ for $i, j \in [K]$ at a particular layer $\ell$. For better visual clarity, we visualize each block by randomly picking 4 directions for each subspace $\mathbf{U}_i$. We observe that the learned $\mathbf{U}^{\ell}_{[K]}$ are indeed approximately incoherent, which aligns well with our assumptions.

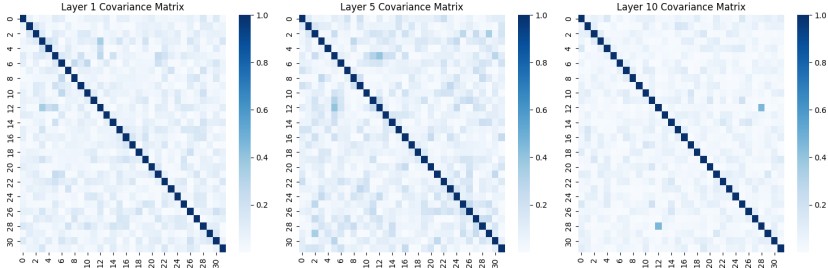

Figure 11: **Visualization of learned subspaces in TSSA blocks at different layers.**

## E PSEUDOCODE OF ToST ARCHITECTURES

See Algorithms 1, 3 and 4 on the following pages.

---

**Algorithm 1:** TSSA Attention Layer of TOST in `PyTorch`

---

```python
class Attention(nn.Module):
    def __init__(self, dim, num_heads = 8, qkv_bias=False):
        self.heads = num_heads
        self.attend = nn.Softmax(dim = 1)
        self.qkv = nn.Linear(dim, dim, bias=qkv_bias)
        self.temp = nn.Parameter(torch.ones(num_heads, 1))
        self.to_out = nn.Sequential(
            nn.Linear(dim, dim)
        )

    def forward(self, x):
        w = rearrange(self.qkv(x), 'b n (h d) -> b h n d', h =
        ↪   self.heads)
        b, h, N, d = w.shape
        w_normed = torch.nn.functional.normalize(w, dim=-2)
        # Denoising against union of Gaussians
        Pi = self.attend(torch.sum(w_normed ** 2, dim=-1) * self.temp)
        dots = torch.matmul((Pi / (Pi.sum(dim=-1, keepdim=True) +
        ↪   1e-8)).unsqueeze(-2), w ** 2)
        attn = 1. / (1 + dots)
        out = - torch.mul(w.mul(Pi.unsqueeze(-1)), attn)
        out = rearrange(out, 'b h n d -> b n (h d)')
        return self.to_out(out)
```

---

**Algorithm 2:** Causal Variant of TSSA Attention in `PyTorch`

---

```python
class Attention(nn.Module):
    def __init__(self, dim, num_heads = 8, max_position_embeddings =
    ↪   1024, qkv_bias=False):
        self.heads = num_heads
        self.attend = nn.Softmax(dim = 1)
        self.qkv = nn.Linear(dim, dim, bias=qkv_bias)
        self.temp = nn.Parameter(torch.ones(num_heads, 1))
        self.to_out = nn.Sequential(
            nn.Linear(dim, dim)
        )
        self.bias = nn.Parameter(torch.zeros(n_head,
        ↪   max_position_embeddings,1))
    def forward(self, x):
        w = rearrange(self.qkv(x), 'b n (h d) -> b h n d', h =
        ↪   self.heads)
        b, h, N, d = w.shape
        w_sq = w ** 2
        w_sq_normed = (w_sq / torch.cumsum(w_sq,dim=-2)) +
        ↪   self.bias[:,:N,:]
        # Denoising against union of Gaussians
        Pi = self.attend(torch.sum(w_sq_normed, dim=-1) * self.temp)
        dots = torch.cumsum(w_sq * Pi.unsqueeze(-1), dim=-2) /
        ↪   (Pi.cumsum(dim=-1)).unsqueeze(-1)
        attn = 1. / (1 + dots)
        out = - torch.mul(w.mul(Pi.unsqueeze(-1)), attn)
        out = rearrange(out, 'b h n d -> b n (h d)')
        return self.to_out(out)
```

---

**Algorithm 3:** MLP Layer of TOST in `PyTorch`

---

```python
class MLP(nn.Module):
    def __init__(self, in_features, hidden_features=None,
    ↪  out_features=None, act_layer=nn.GELU):
        super().__init__()
        out_features = out_features or in_features
        hidden_features = hidden_features or in_features
        self.fc1 = nn.Linear(in_features, hidden_features)
        self.act = act_layer()
        self.fc2 = nn.Linear(hidden_features, out_features)

    def forward(self, x):
        x = self.fc1(x)
        x = self.act(x)
        x = self.fc2(x)
        return x
```

---

**Algorithm 4:** Transformer Block of TOST in `PyTorch`

---

```python
class Transformer(nn.Module):
    def __init__(self, dim, depth, heads, dim_head):
        super().__init__()
        self.layers = nn.ModuleList([])
        self.heads = heads
        self.depth = depth
        self.dim = dim
        for _ in range(depth):
            self.layers.append(nn.ModuleList([
                PreNorm(dim, Attention(dim, heads = heads, dim_head =
                ↪  dim_head)),
                PreNorm(dim, Mlp(in_features=dim,
                ↪  hidden_features=int(dim * 4.0),
                ↪  act_layer=nn.GELU))
            ]))

    def forward(self, x):
        depth = 0
        for attn, ff in self.layers:
            grad_x = attn(x) + x
            x = ff(grad_x) + grad_x
        return x
```

---

## F    LIMITATIONS AND FUTURE DIRECTIONS

Due to limitations in computational resources and time, we have only tested and compared the new architecture up to medium sizes on the ImageNet 1K dataset. Although our ToST architecture enjoys higher computational efficiency, which is increasingly crucial at larger scales, it remains to verify in the future whether its accuracy remains competitive against other architectures in very large scale applications. Finally, the focus of this work is on improving the efficiency and scalability of the attention block of the transformer, so we have kept the MLP block unchanged. Preliminary ablation studies in Table 8 suggest that the vanilla ISTA block used in CRATE might not be optimal for ToST. We leave the study of designing a more effective, efficient, and white-box replacement for the MLP block for future.

