# OpenReview forum: "Token Statistics Transformer: Linear-Time Attention via Variational Rate Reduction"
_ICLR.cc/2025/Conference — ICLR 2025 Spotlight_

### Official Review · Reviewer_VpkN · 2024-10-26

**Soundness:** 4
**Presentation:** 4
**Contribution:** 3
**Rating:** 8
**Confidence:** 4

**Summary:**

This paper introduces Token Statistics Self-Attention (TSSA), a novel attention mechanism derived from a variational form of the Maximal Coding Rate Reduction objective. TSSA exhibits linear computational and memory complexity. Furthermore TSSA significantly departs from traditional pairwise similarity-based attention. By replacing standard self-attention with TSSA in transformer models, the resulting transformer achieves competitive performance on various tasks in terms of expressibility, while maintaining linear complexity.

**Strengths:**

- The authors propose a novel attention mechanism, which is well justified.
- Their attention mechanism has a time and memory complexity of $O(N)$.
- Based on the provided results, their attention mechanism seems to do a better job than standard attention on focusing on the correct pixels.

**Weaknesses:**

- Since the main motivation behind the design is improved time and memory complexity, the comparisons should've been made with FlashAttention. Given that FlashAttention is calculating the softmax-based attention (regular attention) in an efficient manner, and is prevalent (arguably more prevalent the standard attention), it is not fair to compare your time and memory with standard attention. (For reference, FlashAttention has a memory complexity of O(N), and has demonstrated a 2-4$\times$ speedup for context lengths of 4-8K [1]. Therefore, a GPT2 model equipped with FlashAttention would probably outperform your design for the experiment outlined in Table 4-right.)

[1] FlashAttention-2: Faster Attention with Better Parallelism and Work Partitioning

**Questions:**

- I'm curious about the results in Figure 5. Could you provide some intuition on why you attention does such a good job on isolating the attention on the main object? This isolation could result into improved robustness over the standard attention, which is worth investigating.

---

### Official Review · Reviewer_SUwu · 2024-11-01

**Soundness:** 3
**Presentation:** 4
**Contribution:** 3
**Rating:** 8
**Confidence:** 4

**Summary:**

This paper proposes a new attention mechanism that is derived from optimizing general MCR2-like objective. In particular, the derived architecture performs (approximate) low-rank sub-space projection in each attention head. Due to elimination of the classical pairwise attention, the proposed architecture has linear computational complexity. Aside from theoretical derivation, the authors have performed extensive empirical studies to evaluate the performance of the proposed architecture.

**Strengths:**

1- This paper proposes an interesting optimization-based and white-box design perspectives in designing attention-based architectures. I can see the proposed methodology can be used to derive similar attention architectures beyond what has been proposed in this paper, which is an important contribution.

2- The theoretical derivation is sound and quite clear.

3- The empirical studies reported in the paper along with ablation studies are quite comprehensive. It's specially commendable that the authors have covered a variety of tasks and real-world datasets.

4- The paper is very well-written and clear with its flow.

**Weaknesses:**

1- It seems the low-dimensionality of subspace within each head is the key to the performance benefit of the proposed framework. However, such gain may come at some costs that have not been properly discussed in the paper. In particular:

a) There is no guarantee (or soft encouragement) in the proposed framework that the direct sum of the subspaces in each layer should cover the entire (or most of) original space. This can potentially lead to generalization issues specially for out-of-sample test datapoints, where the test datapoints may have small components within the learned subspaces.

b) Fine-tuning might be more challenging as it may require learning new supspaces that are orthogonal to the pre-trained subspaces, which is quite challenging.

To further understand the potential compromises here, I'd recommend authors report further empirical results comparing the zero-shot, out-of-sample performance of TOST with that of regular transformers as well as their convergence rates for fine-tuning.

2- The idea of suppressing less pronounced directions within a set of token vectors may remove discriminating information that might be otherwise crucial to the end task. For instance, in some language tasks, the presence of certain (outlier) words in the input text can be quite decisive for the classification task, but since such token vectors are somewhat outliers, they can be suppressed by the proposed architecture; whereas, in classical pairwise attention mechanism, such intricacies can be organically captured by pairwise attention weights. Since this can pose a significant limitation of the proposed framework for tasks with such characteristics, I'd recommend further empirical studies on such tasks and comparing its results to that of the standard transformers. For instance, one such task can be in-context retrieval for long context sizes in language, e.g. the Needle in A Haystack benchmark. It'd be also quite insightful to visualize rare, yet informative tokens across different layers.

3- Please report the model sizes for all model-dataset pairs in Table 3 (LRA results). It's not clear how much of the difference might be due to having different number of parameters, which wouldn't be a fair comparison.

4- The current work has not been properly placed in related literature. In addition to novel attention operators, I believe the current work is highly related to various formulations of linear attention, but such comparisons are largely missing in the current manuscript. I'd recommend the authors to include a Related Work section as well as adding one or two linear attention baselines (of the same size) in other experiments than LRA.

**Questions:**

Please see comments above.

Additionally, in order to achieve linear complexity, the proposed architecture gets rid of pairwise token interactions and instead creates indirect weak interactions via sample level statistics between tokens. This however comes at a cost of accuracy degradation. This makes me wonder how much such indirect interactions actually contributes to the final accuracy anyway. In particular, what happens if we simply replace $\pi$ with uniform probability distribution? In other words, is there any significant accuracy benefit to indirect interaction vs no interaction at all?

---

### Official Review · Reviewer_zZ2W · 2024-11-03

**Soundness:** 4
**Presentation:** 3
**Contribution:** 3
**Rating:** 8
**Confidence:** 3

**Summary:**

The authors present a mathematically-grounded architecture with operators that resemble the components of the popular transformer objective. Building upon previous works on "white-box" architecture design, specifically one that interprets each network layer as an optimization step of a maximal coding rate reduction (MCR$^2$) objective, and derive a variational form of the objective leading to linear compute and memory requirements. The authors provide detailed experiments on vision, language and long sequence tasks, and also provide a pseudocode implementation and visualize attention maps and estimated memberships at different layers in their proposed network.

**Strengths:**

This is a fundamentally strong work that builds on the concept of white box transformers, specifically one that proposes a MCR$^2$ objective, leading to alternate operators that fundamentally act as attention and mlp blocks in a conventional transformer layer. This work is a great follow-up to it, for reasons listed below -

- The derived variational form of the objective, backed up by sound theory, leads to a linear compute and memory requiring alternative to transformer.
- The approach leads to strongest results among transformer architectures on long sequence tasks and has reasonable performance on vision and language tasks
- The visualisation of attention maps and estimated memberships indicate that minimization of the proposed objective leads to intuitive behaviour of the proposed method.
- The paper, along with its theory and experimental segments, is very well written, and is easy to follow

**Weaknesses:**

- An evident weakness of the method in its current form is the low results on various benchmarks. However, the theoretically sound nature of the network and intuitive visualisations are extremely promising, and follow-up works should be able to extend these ideas to construct more empirically strong networks.

- The attention maps produced by the method seem more coarse and consist of high-norm artifacts, an issue also observed in vision transformers but to a lesser extent. In the case of ViTs, some works have suggested [1] having additional tokens as a remedy to this problem, as certain tokens start behaving as global tokens, leading to high-norm artifacts in the attention maps. It would be good to understand the authors' view on this issue in their approach.

- The results on vision and language tasks do not mention the compute and memory savings, it would be useful to have those, when comparing against other baselines.

[1] Vision Transformers Need Registers, Darcet et. al, https://arxiv.org/abs/2309.16588

**Questions:**

See weaknesses above

---

### Official Review · Reviewer_McCu · 2024-11-04

**Soundness:** 4
**Presentation:** 4
**Contribution:** 2
**Rating:** 6
**Confidence:** 3

**Summary:**

The paper introduces an efficient transformer variant underpinned by a novel linear complexity attention mechanism. The new transformer variant is dubbed the Token Statistics Transformer. The paper builds heavily on previous work, specifically on CRATE (Yu et al., '23), to derive this attention mechanism.

The paper follows a mathematically principled approach, adapted from CRATE, where a high-level data compression objective is optimized by the network's construction, with each network layer corresponding to an optimization step.

The high-level objective introduced in this paper is a novel variational variant of the maximal coding rate reduction objective from CRATE. Optimizing this objective analytically leads to the efficient linear-complexity attention mechanism entitled "token statistics self-attention". This is pretty neat. The authors also include empirical results validating the new transformer variant on relevant tasks in vision & language.

**Strengths:**

- The paper is very well written; the presentation is top notch.
- The fact that the method is actually supported mathematically is quite appealing -- this is becoming rare in the age of deep learning.
- The paper includes experiments on both vision & language modelling tasks, showing that the ToST can be made to work (especially for vision tasks).

**Weaknesses:**

- Experiments on causal language modelling show that the architecture, in its current form, is not very competitive with standard transformer architectures, as indicated by the results in Table 4 -- approximately tripling the number of parameters yields similar performance to GPT2-Base. Looking at Table 4 (right) this would mean that the running time of the ToST is considerably higher than that of GPT-2 when matched for performance -- not exactly re-assuring. Perhaps follow up works can build upon this and find a competitive variant of ToST.

- Experiments on the Long Range Arena are not exactly like-for-like, as the ToST hyper-params are tuned individually for each task -- to my complete surprise. See Table 7 in line 972 in appendix. So, I'm not sure what the takeaway is from Table 3 -- if the method needs tuning for each specific task that definitely takes quite a bit away from its appeal. Transformers are amazing (in part) because they require little tuning from task to task.

- An in-depth section comparing the resulting architecture with state-space models is surprisingly missing. They have similar properties, in terms of compute complexity, but SSMs can attain much better performance in practice.

**Questions:**

See above.

---

### Meta-Review · Area_Chair_2VUR · 2024-12-23

**Metareview:**

Reviewers liked the formulation for the linear attention mechanism proposed in the paper. Main criticism is lack of enough comparisons given the huge amount of related work on Linear attention and SSM style approaches to improve efficiency of Transformers. Also the proposed method seems to have weaker performance on language modeling. Authors promised to add more discussion in the final version. Overall I think the paper is borderline accept and I encourage authors to add more discussions placing the work in the huge literature of linear attention approaches.

**Additional Comments On Reviewer Discussion:**

Authors promised to add more discussion/comparison of related works on linear attention.

---

### Decision · Program_Chairs · 2025-01-22

Accept (Spotlight)